# The future of antibiotic use in livestock

**Alejandro Acosta** [1] ✉, **Wondmagegn Tirkaso** [1], **Francesco Nicolli**[1], **Thomas P. Van Boeckel** [2,3,4], **Giuseppina Cinardi**[1] & **Junxia Song**[1]

Governments worldwide have pledged to reduce antimicrobial use in the agrifood system. This study projects global livestock antibiotic use quantities through 2040 under various scenarios. This work indicates that under a business-as-usual scenario, global antibiotic use could reach ~143,481 tons by 2040, representing a 29.5% increase from the 2019 baseline of ~110,777 tons. However, alternative scenarios suggest that these projections could vary by +14.2% to -56.8%, depending on changes in livestock biomass and antibiotic use intensity. A key contribution of this research is the development of the Livestock Biomass Conversion method, a novel indicator offering improved accuracy in estimating livestock biomass. The findings have important policy implications, highlighting that meaningful reductions in antibiotic use quantity can only be achieved through coordinated efforts targeting both antibiotic use intensity and livestock biomass.

Governments worldwide have endorsed the 79th United Nations General Assembly (UNGA) declaration, committing to a significant reduction in the global quantity of antimicrobials use (AMU) in the agrifood system by 2030[1]. Furthermore, 47 countries have pledged to decrease AMU in agrifood systems by 30-50% by 2030, as outlined in the Muscat Manifesto[2]. Achieving this ambitious target presents significant challenges, particularly in regions where livestock biomass (LBIO) is projected to increase driven by population growth and rising incomes. Understanding the various scenarios under which this goal can be achieved is crucial for shaping effective policy measures and guiding global efforts[3].

Previous studies have projected the future trajectories of antibiotic use quantity (AMUQ) in animal production. Mulchandani et al.[4] estimated that total AMUQ in livestock could reach ~107,472 tons by 2030 [95% CI: 75,972 – 202,661], while Tiseo et al.[5] anticipated a value of ~104,079 tons. These studies employed the population correction unit (PCU) approach to calculate antibiotic use intensity (AMUI), dividing AMUQ (the numerator) by LBIO (the denominator). However, several studies have raised concerns about the PCU indicator, primarily due to the accuracy of the denominator. Bulut and Ivanek[6] pointed out its imprecision in reflecting animal's average weight. Radke[7] criticized its failure to account for animal lifespan. Sanders et al.[8] argued that using the number of slaughter animals as the denominator overlooks the time animals are at risk of antimicrobial treatment. Li et al.[9] highlighted that the lack of subgroup-specific data on liveweight and population dynamics can lead to significant inaccuracies when calculating LBIO across diverse livestock systems. These methodological shortcomings can result in an overestimation of LBIO, and consequently an underestimation of AMUI, typically measured in milligrams of active ingredients per kilogram of LBIO. Accurately estimating LBIO is important not only for refining AMUI-related metrics, but also for evaluating the economic market value of farm animals[10].

This study aims to project the expected global change in livestock AMUQ by 2040, addressing previous methodological limitations through four major contributions. First, we refined existing methods for calculating LBIO[4,11,12] by developing the new Livestock Biomass Conversion (LBC) method. This indicator incorporates detailed live weight data, reflecting differences across animal species, commodity, production systems, production cycles, and cohort levels. Second, we leveraged a unique internal global dataset from the FAO[13] to implement the LBC, enabling more accurate LBIO calculations. Third, we expanded the analytical framework by incorporating a time-series econometric approach to enhance AMUQ foresight analysis. Finally, we explored multiple potential future trajectories for AMUQ in livestock, factoring in the uncertainties in both LBIO growth and AMUI.

In this study, we estimate that global AMUQ in livestock reached approximately ~110,777 tons in 2019, based on the newly

[1]Animal Production and Health Division, Food and Agriculture Organization of the United Nations (FAO), Rome, Italy. [2]One Health Institute, University of Zürich, Zurich, Switzerland. [3]One Health Trust, Washington DC, US. [4]Spatial Ecology and Epidemiology Lab, Université Libre de Bruxelles, Brussels, Belgium. ✉e-mail: alejandro.acosta@fao.org

developed LBC method, compared to -99,414 tons using the PCU method. Our projections suggest that under a business as usual (BAU) scenario, AMUQ could rise to -143,481 tons by 2040 [95% CI: 123,979–163,789], representing a -29.5% increase from the 2019 baseline. However, these projections could vary significantly, with potential deviations ranging from +14.2% to -56.8%, depending on changes in LBIO and AMUI. These findings have important policy implications, highlighting that meaningful reductions in AMUQ can only be achieved through coordinated efforts targeting both AMUI and LBIO.

## Results

### Livestock AMUQ and AMUI in the baseline year (2019)

We analyze the levels of livestock AMUQ and AMUI per region by 2019 using both the LBC and PCU methodologies. This analysis serves two main purposes: first, to establish the level of AMUQ in the baseline year (Fig. 1a), and second, to derive the AMUI parameter for use in the BAU scenario projections (Fig. 1b). We estimated the global AMUQ in livestock for 2019 to be -110,777 tons under the LBC method and -99,414 tons under the PCU method. The LBC method consistently reports higher AMUI compared to the PCU method, with notable differences observed in the North America (-52%), followed by Africa (-40%), Asia and the Pacific (-36%), South America (-13%), and Europe (-12%).

The observed differences arise from the distinct methodologies used to calculate LBIO. The PCU method employs average slaughter weight as a proxy for biomass, leading to lower and less detailed estimates. In contrast, the LBC method integrates comprehensive data, including species-specific live weights, commodity groups, production systems, production cycles, and cohort-level distinctions, offering a more detailed and granular representation of LBIO at global level. Consequently, AMUI values calculated using the LBC method reflect this greater level of detail and accuracy. It is essential to note that AMUI is a derived metric, calculated by dividing total AMUQ by LBIO. While AMUQ remains constant across both methods, differences in LBIO calculations directly impact the resulting AMUI values, highlighting the critical importance of using detailed methodological approaches for global biomass estimation.

### Global livestock AMUQ by 2040

We projected the global AMUQ in livestock by 2040 under a BAU scenario (Fig. 2). This scenario assumes AMUI to continue to follow current trends. Our projections indicate that global AMUQ in livestock

could reach -131,411 tons by 2030 [95% CI: 115,016–148,532] and -143,481 tons by 2040 [95% CI:123,979–163,789]. These figures represent an -18.6% and -29.5% increase, respectively, compared with the baseline year. The confidence intervals (CI) provide upper and lower bounds within which the actual values are expected to fall 95% of the time, reflecting the uncertainty in our projections.

The plot reveals a steady upward trend in livestock AMUQ, marked by a sharp decline around 2018–2019. This drop can likely be attributed to the African Swine Fever (ASF) outbreak, which caused a significant reduction in pig populations, particularly in Asia[14–16]. The model's ability to capture the speed of adjustment following the ASF shock and deviations from the long-term equilibrium is a key feature, highlighting its capacity in accounting for exogenous events.

### Regional livestock AMUQ by 2040

We projected AMUQ in livestock across different regions by 2040 (Fig. 3). Asia and the Pacific is expected to remain the largest contributor, accounting for -64.6% of the global total, with an estimated AMUQ of -92,687 tons [95% CI: 80,932–105,094]. South America is projected to follow, contributing around -19% of the global total, with an estimated -27,197 tons [95% CI: 25,458–29,003]. Africa's AMUQ is forecasted at -8173 tons [95% CI: 7095–9256], making up roughly -5.7% of the global total. North America is projected to account for -5.5%, with an estimated -7922 tons [95% CI: 6986–8860]. Europe is expected to contribute -5.2% to the global total, with a projected AMUQ of -7501 tons [95% CI: 3507–11,575]. It's important to underscore that those regions with the highest AMUQ growth are also those anticipated to play a major role in the global supply of animal-source foods, driven by increasing global food demand by 2040.

### Relative change of livestock AMUQ (2019–2040)

Our analysis reveals significant regional variation in the projected evolution of livestock AMUQ between 2019 and 2040. Figure 4a highlights the relative change in AMUQ across different regions, while Fig. 4b displays the projected growth rates. The results suggest that regions with rapid LBIO expansion will show the most significant increases in AMUQ. Specifically, we expect substantial rises in Asia and the Pacific (-41.1%), Africa (-40.8%) and South America (-19.6%). In contrast, minimal changes are anticipated in Europe (-0.6%), with Northern America expected to see a slight decline (- −3.1%).

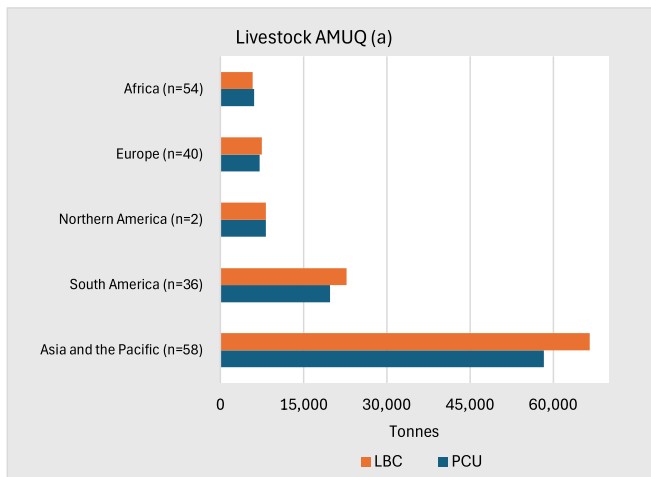

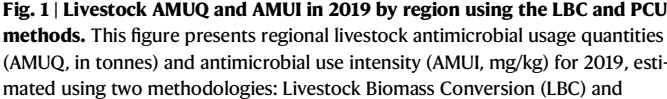

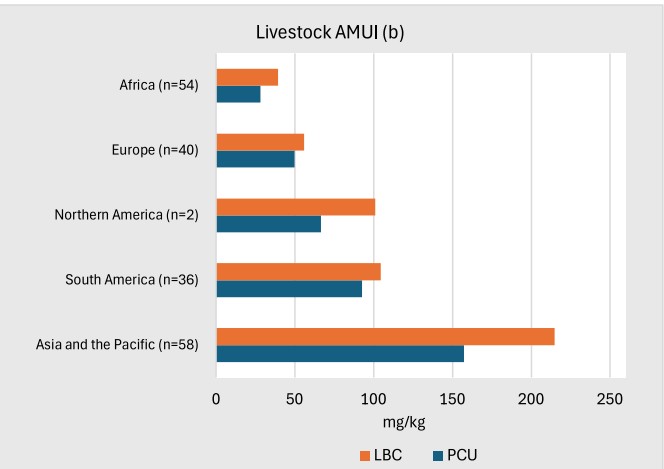

**Fig. 1 | Livestock AMUQ and AMUI in 2019 by region using the LBC and PCU methods.** This figure presents regional livestock antimicrobial usage quantities (AMUQ, in tonnes) and antimicrobial use intensity (AMUI, mg/kg) for 2019, estimated using two methodologies: Livestock Biomass Conversion (LBC) and Population Correction Unit (PCU). Panel (**a**) depicts AMUQ levels across regions, while panel (**b**) illustrates AMUI values. The orange bars represent LBC-based estimates, while the blue bars correspond to PCU-based estimates. The sample size under each region is indicated by *n*.

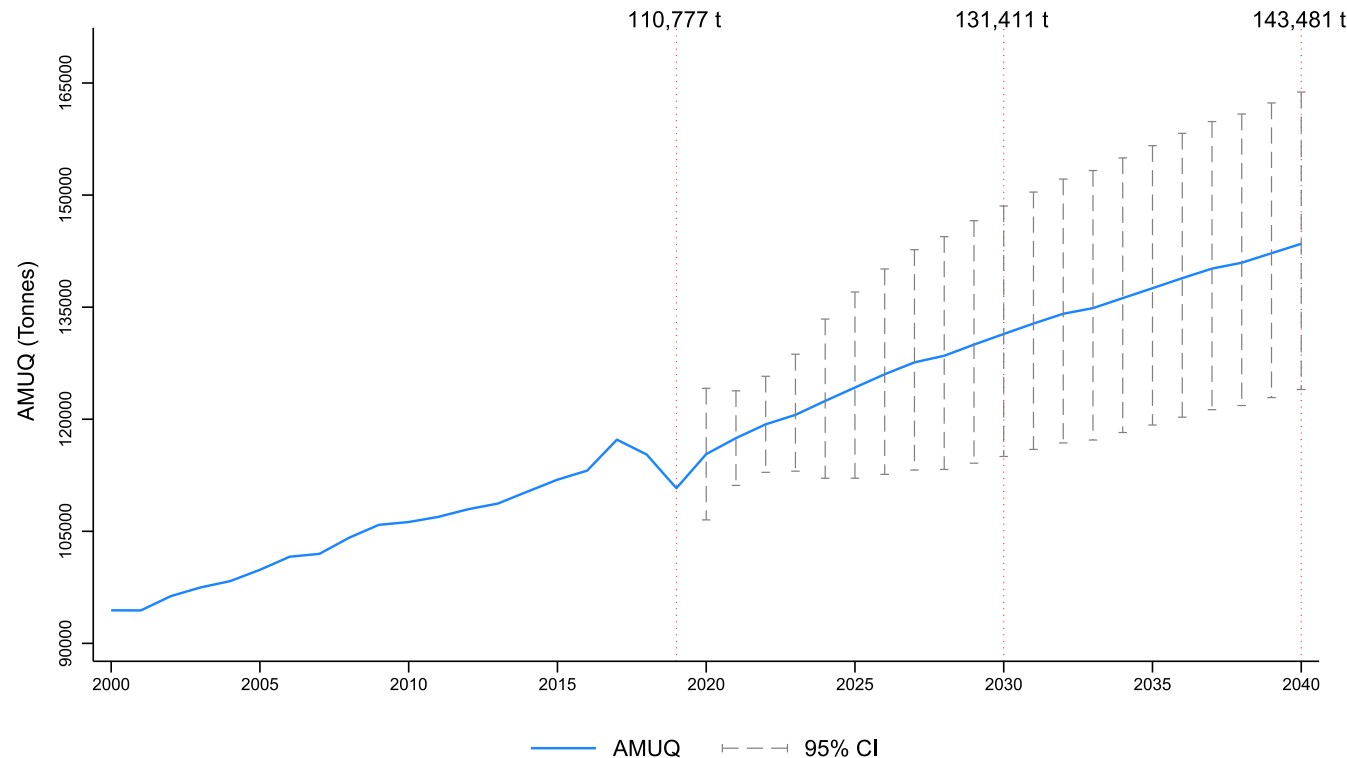

**Fig. 2 | Global livestock AMUQ projections for 2030 and 2040 under the BAU scenario.** This figure presents projected global antimicrobial usage quantity (AMUQ) in tonnes for livestock under a BAU scenario from 2000 to 2040. The solid blue line represents the projected AMUQ trend, while the dashed lines and error bars indicate the 95% confidence intervals over the projection period.

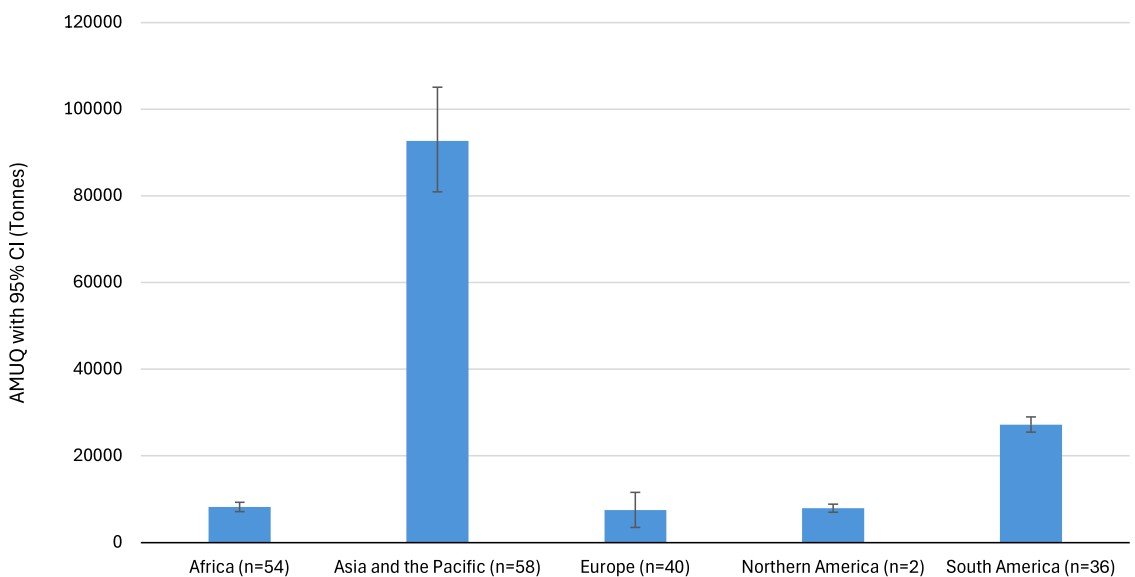

**Fig. 3 | Projected regional livestock AMUQ by 2040 with 95% confidence intervals (CI).** This figure presents the projected antimicrobial usage quantity (AMUQ, in tonnes) in livestock across five global regions by 2040. The projections include 95% confidence intervals (error bars) to indicate the level of uncertainty in the estimates.

The projected growth rates largely mirror these trends. Globally, the annual growth rate in livestock AMUQ is projected to be -0.7%. However, higher growth rates are forecasted for Asia and the Pacific (-1.7%), Africa (-1.6%) and with modest growth expected in South America (-0.9%) and stagnation in Europe (-0.0%). In contrast, North America is projected to experience a decline in AMUQ growth rates (~ −0.1%).

## Livestock AMUQ trajectories by 2040

We explore potential trajectories for livestock AMUQ by 2040 under various scenarios, as outlined in Table 1. These scenarios range from a BAU trajectory, where both LBIO and AMUI remain constant, to various combinations of upper and lower LBIO bounds with AMUI reductions of 30% or 50%. Figure 5 illustrates the percentage deviation in livestock AMUQ from the BAU scenario by 2040 under the eight different

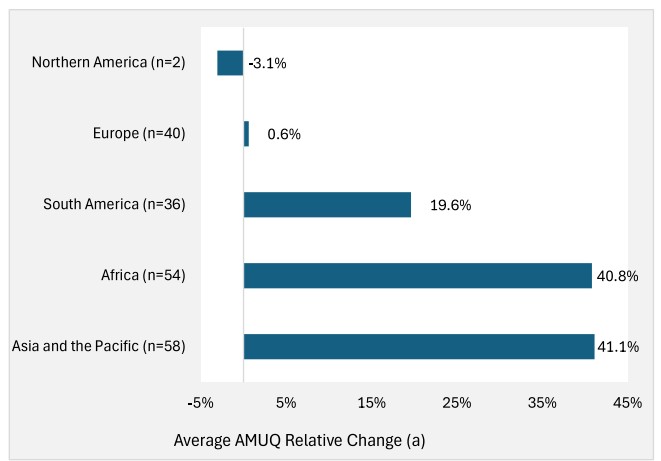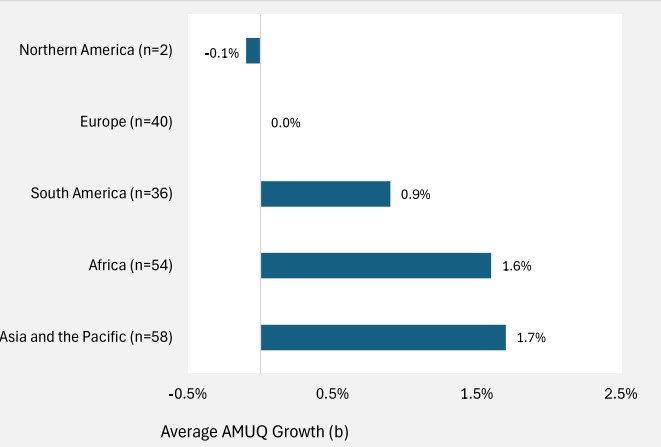

**Fig. 4 | Relative change and growth rates in livestock AMUQ per region (2019–2040).** This figure illustrate regional variation in the projected evolution of livestock antimicrobial usage quantity (AMUQ) between 2019 and 2040. Panel (**a**) presents the relative change (%) in AMUQ, while panel (**b**) shows the average annual growth rates (%) for the same period across different regions.

**Table 1 | Livestock AMUQ trajectories under different LBIO and AMUI scenarios**

| Scenario | Definition | Description |
|---|---|---|
| S1 | LBIO-UP*AMUI-BAU | LBIO at upper bound, AMUI remains at BAU. |
| S2 | LBIO-LOW*AMUI-BAU | LBIO at lower bound, AMUI remains at BAU. |
| S3 | LBIO-UP*AMUI-30% | LBIO at upper bound, AMUI decreases by 30%. |
| S4 | LBIO-BAU*AMUI-30% | LBIO remains at BAU, AMUI decreases by 30%. |
| S5 | LBIO-LOW*AMUI-30% | LBIO at lower bound, AMUI decreases by 30%. |
| S6 | LBIO-UP*AMUI-50% | LBIO at upper bound, AMUI decreases by 50%. |
| S7 | LBIO-BAU*AMUI-50% | LBIO remains at BAU, AMUI decreases by 50%. |
| S8 | LBIO-LOW*AMUI-50% | LBIO at lower bound, AMUI decreases by 50%. |

Scenarios for livestock AMUQ trajectories under different LBIO and AMUI assumptions. This table outlines scenarios for livestock antimicrobial usage quantity (AMUQ) trajectories based on assumptions for livestock biomass (LBIO) and antimicrobial use intensity (AMUI). LBIO is modeled at upper bound, lower bound, or business-as-usual (BAU) levels, while AMUI is considered under BAU and with reductions of 30% and 50%.

scenarios outline in Table 1. Scenario S1, representing upper-bound LBIO with unchanged AMUI, forecasts a ~14.2% increase in AMUQ, emphasizing the risk of rising AMUQ if livestock numbers grow without reducing AMUI. Conversely, Scenario S2, where LBIO is at the lower bound, but AMUI remains unchanged, shows only a ~14% reduction, indicating that reducing livestock numbers alone has a limited effect on AMUQ. Scenarios involving a 30% reduction in AMUI (S3–S5) reveal that even moderate reductions in AMUI can offset AMUQ increases, especially when combined with lower LBIO. The largest reductions are seen in scenarios where AMUI is reduced by 50% (S6–S8). Scenario S8, which combines lower LBIO with a ~50% AMUI reduction, results in a ~56.8% decrease in AMU.

Figure 6 provides a detailed view of the projected livestock AMUQ trajectories by 2040 across the eight scenarios defined (Table 1). Focusing first on scenarios where livestock biomass remains at the BAU level provides valuable insights into the role of AMUI in determining future AMUQ levels. In the pure BAU scenario, where both LBIO and AMUI continue to follow their current trends, AMUQ is projected to rise from ~110,777 tons in 2019 to ~143,481 tons by 2040. This serves as the baseline scenario, highlighting the likely increase in AMUQ if no interventions are implemented to alter these trends.

In S4, where LBIO remains at BAU levels, but AMUI is reduced by 30%, AMUQ is projected to decrease to ~100,437 tons by 2040. This significant reduction suggests that even moderate cuts in AMUI can substantially offset the AMUQ increase driven by steady LBIO levels. Scenario S7, where LBIO remains unchanged while AMUI is halved, projects an even more dramatic reduction, with AMUQ expected to drop to ~71,741 tons by 2040. Finally, S8, which combines the lowest

LBIO with a 50% reduction in AMUI, achieves the most substantial decrease in AMU, with projections dropping to ~61,989 tons by 2040. This scenario underscores the synergistic effect of targeting both LBIO and AMUI, illustrating that comprehensive strategies addressing both factors are essential for minimizing AMUQ.

**Livestock AMUQ projections across livestock species: A robustness check**

Given that our AMUQ projections rely on AMUI data derived from AMUQ sales data, it is critical to assess the uncertainties and potential biases inherent in this approach. A significant concern is whether the AMUI parameters, which are based on AMUQ sales data, accurately reflect the actual usage of antibiotics in livestock. This is particularly challenging because AMUQ sales data often lack essential details about the characteristics of the treated animal populations—such as age, weight at treatment, health status, or the specific availability and potency of antibiotics. Moreover, sales data may be subject to local reporting biases, which can skew the results, leading to potential over- or underestimation of AMUQ. To address these concerns and uncertainties, we conducted a robustness check of the projected livestock AMUQ by 2040 using an alternative methodological approach. This approach integrates LBIO projections with average AMUI global parameters for different animal species obtained from the literature[4,17]. This alternative BAU scenario allows us to assess AMUQ projections from a species-focused perspective, offering a complementary view to our global AMUQ projection estimates.

Figure 7 illustrates the livestock AMUQ projections robustness check across selected animal species by 2040. The projections show

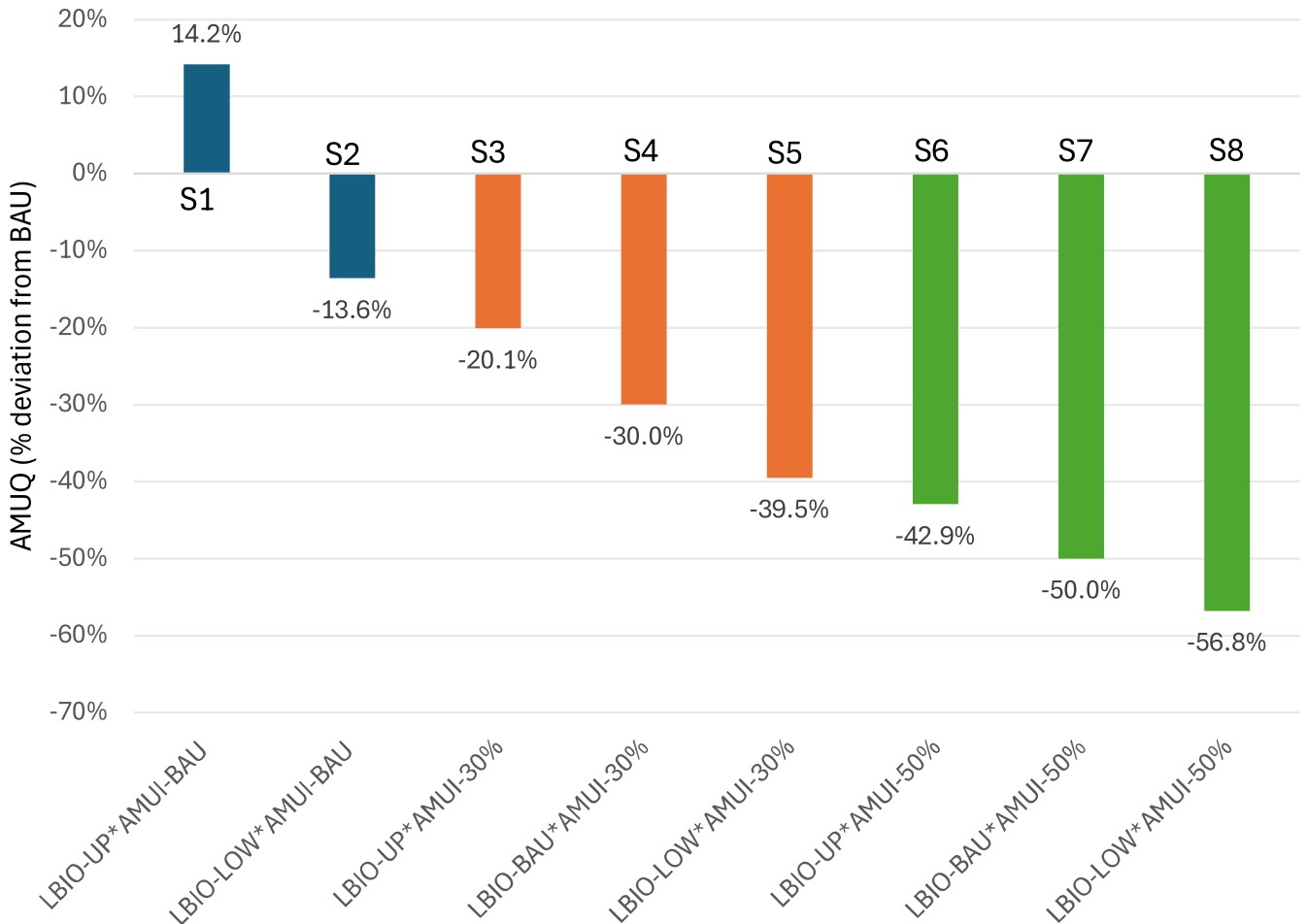

**Fig. 5 | Projected livestock AMUQ by 2040 under different scenarios.** This figure illustrates the percentage deviation in livestock antimicrobial usage quantity (AMUQ) from the business-as-usual (BAU) scenario by 2040 under eight different combinations of livestock biomass (LBIO) and antimicrobial use intensity (AMUI) scenarios. These scenarios reflect varying assumptions about LBIO (upper bound, lower bound, or BAU) and AMUI reductions (none, 30%, or 50%). The scenarios are detailed in Table 1.

a sharp increase in total livestock AMUQ, expected to reach ~120,138 tons by 2040, which marks a ~30% increase from the ~92,151 tons in 2019. This rise is primarily driven by the growth in LBIO. Pigs are projected to remain the dominant consumers of antibiotics, accounting for ~40,646 tons, or nearly ~34% of the total AMUQ by 2040. Cattle are expected to be the second-largest contributors, with an estimated ~31,655 tons of AMUQ. Although sheep and goats have smaller biomass compared to pigs and cattle, they are still projected to consume ~14,441 tons and ~10,682 tons, respectively. Chickens, despite their smaller individual biomass, will account for ~5066 tons of AMUQ by 2040, reflecting the high-intensity of poultry production systems. Meanwhile, aquaculture is projected to experience rapid growth in antibiotic use. By 2040, the sector is expected to consume ~17,648 tons, underscoring the expanding role of farmed fish.

## Discussion

Previous studies, including those by Ardakani et al.,[18] Mulchandani et al.,[4] Tiseo et al.,[5] and Van Boeckel et al.,[19] have projected future antibiotic usage quantities (AMUQ) in livestock populations. This study builds upon those efforts by introducing the Livestock Biomass Conversion (LBC) method, a novel to estimating livestock biomass (LBIO) on a global scale. The LBC method represents a significant advancement over traditional methodologies, such as the Population Correction Unit (PCU), by incorporating detailed live weight data that reflects variations across animal species, commodities, production

systems, production cycles, and cohort levels (Supplementary Fig. A1). A comparison of projections for LBIO and AMUQ using the LBC method (Supplementary Fig. A2) reveals that the projected increase in AMU aligns proportionally with the growth in LBIO.

Our analysis estimates global AMUQ in livestock at ~110,777 tons in 2019 using the LBC method, compared to ~99,414 tons with the PCU method. These result contrasts with WOAH's[20] seventh annual report, which reported a total AMUQ of ~88,987 tons based on data from 94 countries. The discrepancy between the LBC and PCU methods underscores the critical importance of denominator selection in AMUQ and AMUI. As highlighted by Radke[7], Bulut and Ivanek[6], Sander et al.[8] the PCU method has significant limitations, failing to account for variations in animal lifecycles and live weight, which can result in systematic underestimations of AMUI.

Under a BAU scenario, global livestock AMUQ is projected to rise to ~131,411 tons by 2030 (an ~18.6% increase from 2019) and ~143,481 tons by 2040 (a ~29.5% increase). While these trends align with previous studies[4], our projections reveal a steeper rise, largely due to differences in methodologies and baseline assumptions. For instance, Mulchandani's[4] study relied on parameters from the FAO's "Future of Food and Agriculture to 2050" (FOFA) report, which uses 2012 as a baseline and a global partial equilibrium model. In contrast, our analysis utilizes time-series data spanning 1961 to 2021 and applies econometric models for projections covering 2020–2040.

The findings highlight significant regional disparities in livestock AMUQ. By 2040, Asia is projected to remain the largest contributor,

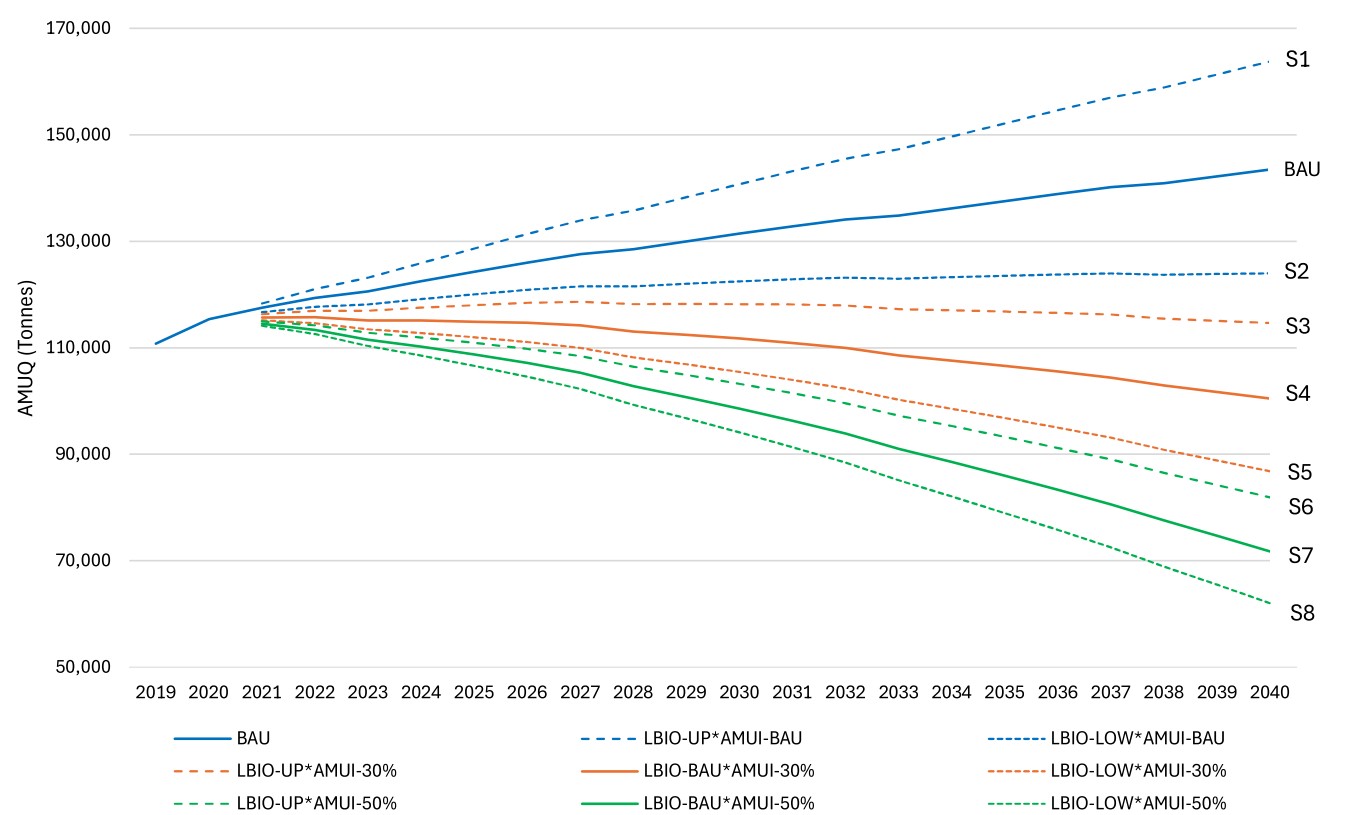

**Fig. 6 | Livestock AMUQ trajectories by 2040 under different scenarios.** This figure shows the projected antimicrobial usage quantity (AMUQ, in tonnes) in livestock from 2019 to 2040 under eight scenarios outlined in Table 1, reflecting varying assumptions regarding livestock biomass (LBIO) and antimicrobial use intensity (AMUI). The bold lines represent the BAU (Business-as-Usual) scenario, as well as Scenario (S4), which assumes a 30% reduction in AMUI while keeping LBIO at the BAU level, and Scenario (S7), which assumes a 50% reduction in AMUI while maintaining LBIO at the BAU level. These scenarios illustrate the impact of AMUI reduction on AMUQ, independent of changes in livestock biomass.

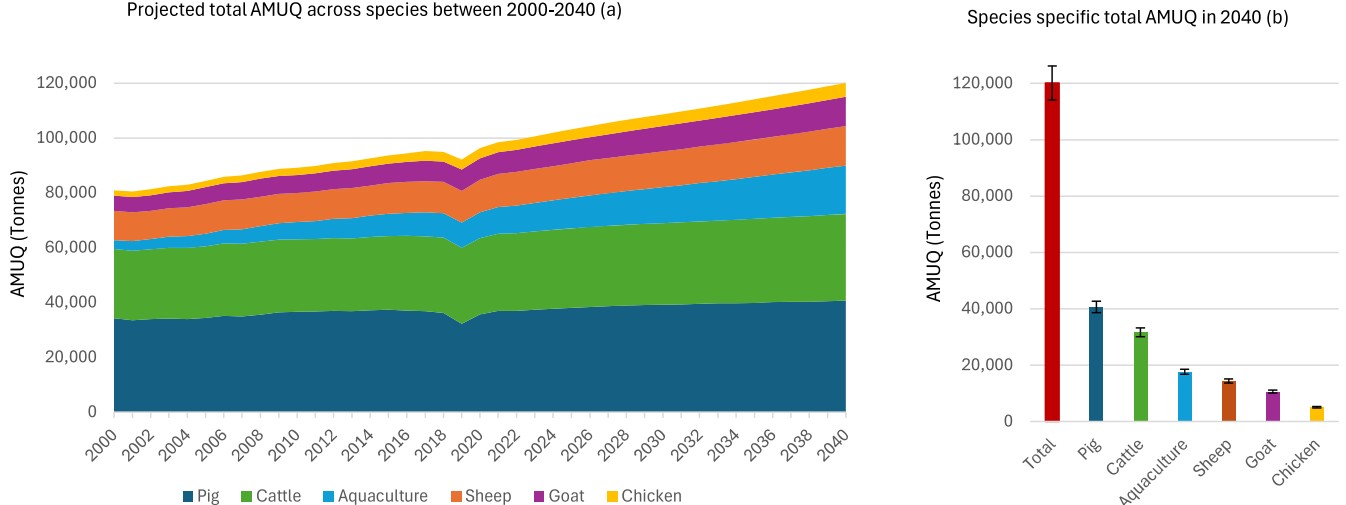

**Fig. 7 | Robustness check of livestock AMUQ projections across animal species by 2040.** This figure presents a robustness check of livestock antimicrobial usage quantity (AMUQ) projections across animal species. Panel (**a**) illustrates historical and projected trends in total AMUQ across all species from 2000 to 2040, while panel (**b**) highlights species-specific contributions to AMUQ in 2040, including 95% confidence intervals (error bars).

accounting for approximately ~63.3% of global livestock AMUQ. This is consistent with OECD findings, particularly regarding Asian countries' intensive use of antibiotics in pig and poultry production sectors[19]. In contrast, Africa is expected to experience the highest relative growth, with AMUQ projected to increase by ~40.8% between 2019 and 2040. Meanwhile, Europe and North America are anticipated to see only minimal increases, largely due to slower LBIO growth and stricter regulations. Importantly, the regions with the highest projected rises in AMUQ coincide with those leading the fastest growth in animal-sourced food production, driven by population growth and rising incomes.

The analysis highlights that achieving significant reductions in AMUQ requires addressing simultaneously reductions in AMUI and the

optimization of LBIO. Optimizing LBIO involves enhancing livestock efficiency through improved management practices, biosecurity, and technological change, with a focus on increasing productivity per animal rather than expanding herd sizes. This integrated approach is essential for managing the complex interplay between AMUQ, LBIO, and the rising global demand for animal-source proteins. Such strategies are especially critical in middle- and low-income countries, where livestock systems play a crucial role in livelihoods, food security, and economic resilience.

The accuracy of our AMUQ projections depends heavily on the precision of baseline AMUQ and AMUI estimates and is influenced by various factors, including global commitments to reduction, national and regional policies, regulatory frameworks, economic incentives, animal health and disease prevalence, production system intensification, and farmers' behaviors toward antibiotic use (Homes et al.[21]; Henriksson et al.[22]). External factors such as consumer preferences, access to vaccines and non-antibiotic alternatives, and environmental conditions also play a critical role in shaping future AMUQ trends (Lhermie et al.[23]; Laxminarayan et al.[24]).

While national policies and farm-level technical interventions are vital, addressing AMUQ as a global challenge requires innovative economic and financial mechanisms to accelerate progress. Mechanisms such as abatement markets, bonds, credits, and a global fund for reduction could provide the incentives needed to drive collective action. These mechanisms would deliver the financial and structural momentum necessary to achieve substantial and sustained reductions in AMUQ, fostering global progress in combating AMR.

Finally, the study underscores that meaningful reductions in AMUQ require coordinated, integrated strategies. To achieve this, a robust, independent mechanism is essential to unify global efforts and ensure consistent progress. The establishment of an independent panel for action against AMR, as recommended by the 79th UNGA declaration[1], could play a critical role in addressing this need. Such a panel would serve as a mechanism to facilitate the generation and use of scientific evidence to tackle AMR effectively. In doing so, it could empower countries to accelerate the implementation of global strategic initiatives, such as FAO's RENOFARM program, which aims to reduce the need of antimicrobials on farms (FAO[25]).

This study provides valuable insights into AMUQ in livestock, but several limitations must be acknowledged. A key constraint is the availability and quality of disaggregated public data on AMUQ. The Antimicrobial Use (ANIMUSE) dataset used in our analysis provides regionally aggregate data on livestock AMUQ across 108 countries. While this represents a substantial dataset, it still has significant gaps. One notable issue is that certain countries do not report data. As a result, we had to employ modeling adjustments to estimate livestock AMUQ and AMUI at regional, and global levels. This adjustment introduces uncertainty, as these estimates are heavily dependent on the assumptions and methodologies applied. For example, extrapolating AMUI levels from reporting to non-reporting countries within a subregion may not capture the specific practices and conditions of the non-reporting nations.

Another limitation is the lack of species-specific data. AMUI varies significantly among animal species and production systems. For instance, poultry production typically involves higher AMUI than cattle. Similarly, the dataset does not disaggregate AMUQ by antibiotic class, which limits the analysis of how specific classes, such as fluoroquinolones and macrolides, contribute to AMUQ trends and resistance patterns. This gap is particularly significant given the global prioritization of these antibiotic classes due to their critical roles in both human and veterinary medicine and their association with resistance risks[26,27]. The reliance on sales data as a proxy for actual AMUQ adds another layer of complexity. Sales data often differ from real usage patterns due to factors such as stockpiling, wastage, and unrecorded transactions. Additionally, sales data lack critical details on species, diseases treated, and unauthorized use, risking underestimation or misrepresentation of AMUQ. These limitations necessitate caution when interpreting trends, particularly in the context of increasing LBIO.

Addressing these limitations requires enhanced data collection methods and standardized reporting practices to improve the accuracy and reliability of AMUQ estimates. Expanding data coverage, particularly in non-reporting countries with significant livestock populations, is crucial to reducing biases and inaccuracies in global projections. This study introduces the LBC method, which offers a more precise assessment of LBIO by incorporating species-specific live weights, production systems, and cohort-level data, addressing key limitations of the traditional PCU approach and improving the denominator for AMUI calculations. While ANIMUSE relies on sales data that may not fully align with actual usage patterns, it remains the primary internationally recognized dataset for tracking global AMUQ.

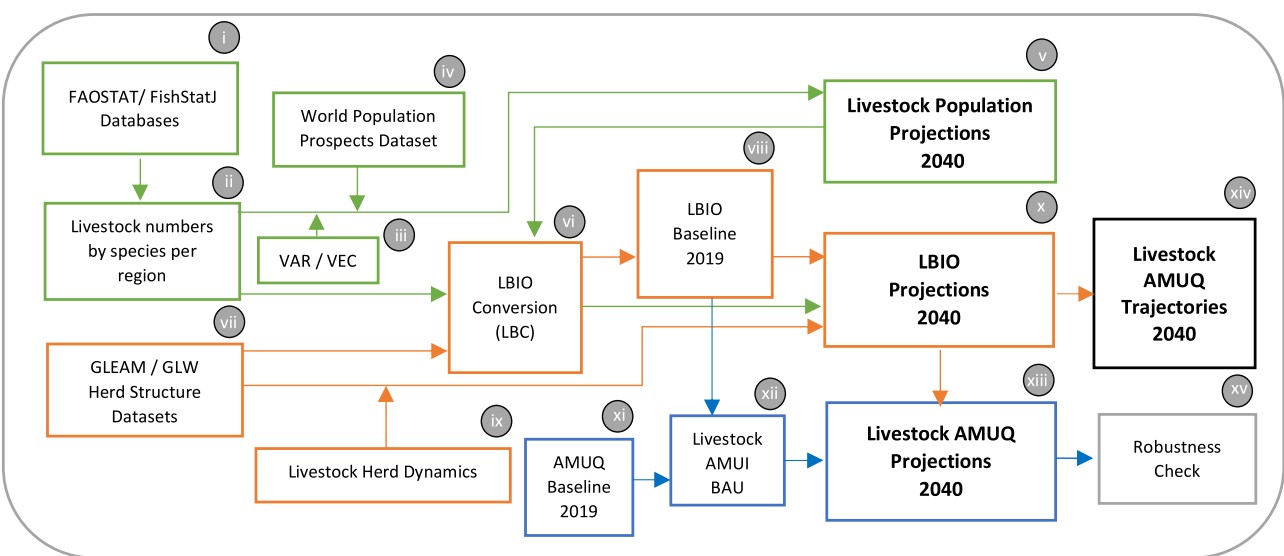

**Fig. 8 | Methodological Framework for Livestock AMUQ Projections.** This figure outlines the methodological framework employed in the study, detailing the integrated steps and datasets used to project livestock antimicrobial usage quantity (AMUQ) by 2040.

## Methods

This section describes the methodological framework employed in this study to project AMUQ by 2040. The framework consists of four key integrated outputs, each involving several steps (Fig. 8). These outputs include projections of the livestock population, LBIO, livestock AMUQ, and potential trajectories for AMUQ.

### Livestock population projections

The first stage involves projecting livestock populations for 2040 **(v)**. This includes using datasets **(i)** from FAOSTAT[28] for terrestrial animals (cattle, pigs, goats, sheep, chickens) and FishStatJ[29] for aquatic species (salmon, catfish, shrimp, tilapia, trout, carp, barramundi, seabass, grouper, milkfish), focusing on species most associated with antibiotic use[17,30,31]. Livestock and aquaculture numbers are disaggregated by region (Africa, Asia and the Pacific, Europe, South America and Northern America) for geographically specific analysis **(ii)**.

We projected the future of livestock populations by 2040 for major animal species (cattle, pigs, goats, sheep, chickens, and selected aquaculture species). This component relies on time series data from 1961 to 2022 and uses vector autoregressive (VAR) and vector error correction (VEC) econometric models **(iii)**. These models are commonly used to examine the long-term and short-term dynamics of time-series information by considering lagged endogenous and exogenous variables. Human population growth projections **(iv)**, sourced from the World Population Prospect (2022), are included as an exogenous variable to account for the demand-driven changes in animal-source food consumption. The modeling procedure is guided by the following steps:

#### Step 1

We determined whether the series are stationary or non-stationary. Therefore, we tested for unit roots using the augmented Dickey–Fuller and Kwiatkowski–Phillips–Schmidt–Shin tests[32,33]. These tests guided the selection of a model that aligned with the statistical properties of the underlying data generation process.

#### Step 2

If the series is stationary, we process the specification of a multivariate vector autoregressive (VAR) model as in Eq. (1), where $y_t$ is a vector of endogenous variables in logarithms, $\Gamma$ is a parameter matrix, $C$ is the vector of coefficients for the constant/trend terms ($D_t$), $\beta$ is the parameter matrix of the exogenous term ($z_t$), and ($v_t$) is a vector of error terms. The specification outlined in Eq. (1) is the basis for generating forecasts for outcome variable ($y_t$) within the system. We assumed that $E(v_t) = 0$, $E(v_t v_t') \neq 0$, and $E(v_t v_s') = 0$ for all $t \neq s$. Consequently, we used the following forecast equation to predict $\hat{y}_t$ as (2) where $\hat{y}_s$, $s < t$, is the estimate of $y_t$ from period $s$ in the forecast horizon.

$$y_t = \Gamma y_{t-1} + \cdots + \Gamma_{p-1} y_{t-p+1} + CD_t + \beta z_t + v_t \tag{1}$$

$$\hat{y}_t = \hat{\Gamma} y_{t-1} + \cdots + \hat{\Gamma}_{p-1} \hat{y}_{t-p} + \hat{C} D_t + \hat{\beta} z_t \tag{2}$$

#### Step 3

If the series are nonstationary, we tested for cointegration using the Johansen Trace Test[34,35]. If the series was cointegrated, we specified a vector error correction model as in Eq. (3), where the $\Gamma$ represents the short-term parameters, $\Pi$ is a matrix of coefficients for the lagged differences of the variables, and $\beta$ is the long-term parameter that contains the cointegration relationship. Based on Eq. (3), we established a framework for generating forecasts for the outcome variable

($\Delta y_t$) following Eq. (4).

$$\Delta y_t = \Pi y_{t-1} + \Gamma \Delta y_{t-1} + \cdots + \Gamma_{p-1} \Delta y_{t-p+1} + CD_t + \beta z_t + v_t \tag{3}$$

$$\Delta \hat{y}_t = \hat{\Pi} \hat{y}_{t-1} + \hat{\Gamma} \Delta \hat{y}_{t-1} + \cdots + \hat{\Gamma}_{p-1} \Delta \hat{y}_{t-p+1} + \hat{C} D_t + \hat{\beta} z_t \tag{4}$$

#### Step 4

If the series are nonstationary and do not exhibit cointegration, the cointegration term, $\Pi y_{t-1}$, in Eq. (3) becomes irrelevant, and the relationship among the endogenous variables ($\Delta y_t$) can be effectively represented by a stable VAR model in different forms. Therefore, a VAR model with $\Delta y_t$ being a vector of endogenous variables can be specified as (5). We further specify the forecast model for Eq. (5) to generate the future outcome value ($\Delta \hat{y}_t$) using Eq. (6).

$$\Delta y_t = \Gamma \Delta y_{t-1} + \cdots + \Gamma_{p-1} \Delta y_{t-p+1} + CD_t + \beta z_t + v_t \tag{5}$$

$$\Delta \hat{y}_t = \hat{\Gamma} \Delta \hat{y}_{t-1} + \cdots + \hat{\Gamma}_{p-1} \Delta \hat{y}_{t-p+1} + \hat{C} D_t + \hat{\beta} z_t \tag{6}$$

#### Step 5

Finally, we present 95% confidence intervals (CI) to reflect the level of uncertainty of our projections, providing the upper and lower bounds within which the actual values are expected to fall 95% of the time. These CIs are estimated by analyzing the residuals of the model and comparing the fitted values with the actual historical values. This analysis helps estimate the variance and covariances of the residuals, which in turn are used to calculate the forecast error variance. The CIs are then constructed using the point forecasts and the standard errors derived from the forecast error variances.

### Livestock biomass conversion (LBC)

The second stage involved the development of the LBC method **(vi)**. The LBC was developed to account for species, commodity type, production systems, production cycles, and live weight variations by cohort. LBIO is defined as the aggregated average weight of food-producing animals each year.

Several methodologies exist for calculating the average biomass of livestock species in AMUQ analyzes, such as those employed by European Medicines Agency (E.M.A)[36] and Canadian Integrated Program for Antimicrobial Resistance (CIPARS)[37]. One widely used approach is the PCU, which normalizes AMUQ data by accounting for both animal population sizes and their average weights, facilitating comparisons across species and regions.

The PCU is determined by multiplying the total number of animal species ($l_i$) in each country by its average live weight at the time of treatment, $w_i$, as Eq. (7). This equation was further standardized to account for the differences in animal weight and the number of production cycles as in Eq. (8), where $j$ denotes the production system of a country, $n_{i,j}$ is the number of production cycles for each species ($i$) in each production system ($j$). $Q_i$ represents the total quantity of meat in each country for each animal species ($i$), and $R_i$ denotes the carcass weight-to-live weight ratio for each animal species ($i$). However, the PCU, as described in Eq. (8), relies on a single average carcass/slaughtered weight of animal species ($i$) in each production system ($j$). Therefore, this approach may result in skewed PCU measurements because it does not consider the weight of each animal at the specific cohort level. In other words, it assumes all animals to be adults.

$$PCU_i = l_i \times w_i \tag{7}$$

$$PCU_{i,j} = I_{i,j} \times n_{i,j} \times \left(\frac{Q_i}{R_k}\right) \tag{8}$$

To overcome this limitation, we introduced a new metric: the livestock biomass conversion (LBC) method. The LBC method accounts for variations in LBIO by incorporating species-specific live weights, differentiated by commodities, production systems, production cycles, and cohort. This calculation is performed following Eq. (9), where $LBC_{ic}$ represents the LBIO indicator for animal species ($i$) in cohort ($c$); $A_i$ is the total number of animal species ($i$); $n_i$ denotes the number of production cycles used to account for lifespan differences among species ($i$), $L_{k,i}$ is the number of livestock species ($i$) under commodity group ($k$); $L_{p,i}$ is the number of livestock species ($i$) under commodity group ($k$) and production system ($p$); $L_{c,i}$ is the number of livestock species ($i$) under commodity group ($k$), production system ($p$), and cohort ($c$); and $W_{c,i}$ is the cohort weight for animal species ($i$) under commodity group ($k$) and production system ($p$).

$$LBC_{ic} = A_i \times n_i \times \left(\frac{L_{i,k}}{A_i}\right) \times \left(\frac{L_{i,p}}{L_{i,k}}\right) \times \left(\frac{L_{i,c}}{L_{i,p}}\right) \times W_{i,c} \tag{9}$$

$$LBC_i = \sum_{c=1}^{n} LBC_{ic} = \sum_{c=1}^{n}\left( A_i \times n_i \times \left(\frac{L_{i,k}}{A_i}\right) \times \left(\frac{L_{i,p}}{L_{i,k}}\right) \times \left(\frac{L_{i,c}}{L_{i,p}}\right) \times W_{i,c} \right) \tag{10}$$

The LBC for animal species ($i$), accounting for all cohorts, is calculated by taking the linear summation of Eq. (10), where $LBC_i$ represents the livestock biomass indicator for animal species ($i$), and $n$ denotes the cohort number.

### Livestock biomass projections
The third step is about the development of LBIO projections (**x**) by linking the estimated livestock populations to their respective biomass. Using herd structure data from FAO's Global Livestock Environmental Assessment Model (GLEAM)[13] and Gridded Livestock of the World (GLW)[38] datasets (**vii**), the LBIO baseline for 2019 was established (**viii**). To project LBIO by 2040, herd structure dynamics (**ix**) were introduced into the GLEAM model, linking production system changes to gross domestic product (GDP) growth as a proxy for economic trends. These herd dynamics were integrated with livestock population projections, and the LBC method was reapplied to produce the LBIO projection for 2040.

### Antibiotic use intensity (AMUI)
The fourth step involves estimating the level of AMUI (**xii**) for each region under a BAU scenario. In this study, AMUI represents the relative amount of AMUQ per unit of LBIO. This study used the 2019 AMUQ data from WOAH's seventh annual report on antimicrobial agents intended for use in animals[20]. In addition, we integrated 2019 antimicrobial sales and distribution data from the United States Food and Drug Administration (U.S. FDA)[39] and Canada's CIPARS[40] to improve the regional disaggregation of results. To estimate AMUI under the BAU scenario, we rely on the 2019 baseline values for AMUQ (**xi**) and LBIO (**viii**). Since the AMUQ data published by WOAH only covers reporting countries, adjustments were made to account for AMUI in regions with non-reporting countries. The following steps outline the methodology for extrapolating AMUI data to account for non-reporting countries:

### Step 1: Estimation of LBIO for reporting countries
The first step is to estimate the level of LBIO for reporting countries in each region. This is necessary because AMUQ data is only available for these countries, and we need to extrapolate to the entire region, including non-reporting countries. The total LBIO for the group of reporting countries $c$, in region $r$, is calculated by multiplying the overall

LBIO in region $r$ by the share of group $c$ SLBIO relative to the total LBIO in that region. This is represented in Eq. (11), where $LBIO_{cr}$ represents the livestock biomass for the group of countries $c$ in region $r$, $LBIO_r$ denotes the total livestock biomass in region $r$, and $SLBIO_c$ is the share of livestock biomass from group $c$ relative to the regional total.

$$LBIO_{cr} = LBIO_r \times SLBIO_c \tag{11}$$

### Step 2: Calculation of AMUI for reporting countries
Once the LBIO for reporting countries is estimated, we can calculate the level of AMUI for these countries. To calculate AMUI in region $r$, we divide AMUQ for the group of countries $c$ in region $r$ by the corresponding LBIO for that group. This provides an intensity measure that reflects the amount of AMUQ per unit of LBIO. This is expressed in Eq. (12), where $AMUI_r$ represents the level of livestock AMUI in region $r$, $AMUQ_{cr}$ is the quantity of AMU for the group of countries $c$ in region $r$, and $LBIO_{cr}$ is the corresponding livestock biomass of countries $c$ in region $r$.

$$AMUI_r = \frac{AMUQ_{cr}}{LBIO_{cr}} \tag{12}$$

### Step 3: Extrapolation of AMUI to non-reporting countries
To estimate the total livestock AMUI for the entire region r (including non-reporting countries), we assume that the AMUI calculated for the reporting countries can be applied to the non-reporting countries as well. In other words, we assume that non-reporting countries have a similar AMUI usage per unit of LBIO as the reporting countries. The total AMUI for region $r$, is then calculated by multiplying the total LBIO in region $r$ (including both reporting and non − reporting countries) by the livestock AMUI in the same region. This calculation provides an overall estimate of the level of livestock AMUI for the region. This is express in Eq. (13), where $AMUI_r$ denotes the estimated livestock AMUI in subregion $r$, $LBIO_{cr}$ is the total livestock biomass in region $r$, and $LAMUI_r$ is the livestock AMUI for region $r$.

$$AMUI_r = LBIO_{cr} \times AMUI_r \tag{13}$$

It is important to note that changes in LBIO calculation methods, such as shifting from PCU-based to LBC-based estimates, can influence the resulting AMUI because the calculation is directly tied to the accuracy and granularity of the LBIO estimate. This sensitivity highlights the importance of choosing detailed methods for biomass estimation.

### Livestock AMUQ projections
The fourth step of our analysis involves projecting AMUQ for 2040 (**xiii**), which requires the calculation of AMUI (**xii**) under a BAU scenario. In this study, AMUI represents the relative amount of AMUQ per unit of livestock biomass (LBIO) and is calculated for each region.

In the fourth stage, livestock AMUQ projections for 2040 (**xiii**) are calculated using LBIO projections and AMUI levels under a BAU scenario. AMUI parameters (**xii**) are derived by combining AMUQ data (**xi**) from WOAH's seventh annual report on antimicrobial agents intended for use in animals[20] with LBIO data from the LBC model. This data includes sales and import figures for AMUQ, collected through surveys involving 157 countries, with 121 providing quantitative information, representing around 85% of WOAH's members and 70% of the global animal biomass.

The projection of total AMUQ across regions relies on linking total LBIO projections (**x**) and AMUI (**xii**). We used this relationship to compute the total AMUQ in region $r$ at time $t$ as in Eq. (14), where $AMUQ_{rt}$ denotes the total AMUQ in region $r$ at time $t$, $LBIO_{cit}$ represents the total livestock biomass in country $i$ at time $t$, and $AMUI_r$ is the BAU average regional-level AMUI (milligrams of antibiotic/kilograms

of biomass). We used the latest information on AMUI, given that few countries reported antimicrobial use in the previous years, showing a lack of representativeness. In the BAU scenario, we assumed that the observed AMUI from 2019 would remain unchanged throughout the forecast horizon. Thus, the parameter did not necessarily imply that the baseline figure was optimal. However, it was a reference point for understanding long-term trends without significant changes or interventions.

$$AMUQ_{rt} = \sum_{i=1}^{n} LBIO_{cit} \times AMUI_r \qquad (14)$$

### Livestock AMUQ potential trajectories

Predicting how social, economic, and environmental drivers will influence AMUQ and AMUI trajectories is inherently complex. In our BAU scenario, we assume that socioeconomic factors like population growth and rising incomes will continue as projected, increasing demand for animal protein and driving LBIO growth. We also expect livestock production systems to maintain current productivity trends, with gradual improvements over time. Regarding environmental factors, the BAU scenario assumes that climate change effects will gradually intensify but will not significantly disrupt global livestock production by 2040. Livestock emissions are projected to continue along current trajectories, with only limited regulatory interventions aimed at reducing greenhouse gas emissions.

However, we recognize that these assumptions may be optimistic. Climate change and policy interventions are likely to influence AMUQ pathways. Stricter emissions regulations could constrain the anticipated growth in livestock biomass, while tighter antibiotic regulations could result in substantial reductions in AMUI. For instance, the OECD's 2019 report on AMU in Brazil outlines the implications of AMU regulatory efforts in the livestock sector[41]. To account for these uncertainties, we simulate livestock AMUQ potential trajectories by 2040 **(xiv)** under different assumptions (Table 1).

These scenarios include upper and lower bounds for LBIO projections, based on 95% confidence intervals, reflecting not only population and income growth but also the potential impact of climate conditions and environmental regulations on the livestock sector. Additionally, we simulate AMUI reduction targets of 30% and 50%, aligned with the goals of the Muscat Manifesto adopted at the 2022 Global High-Level Ministerial Conference on AMR[2] in Oman.

### Robustness checks on AMUQ projections

We conducted a robustness check **(xv)** using an alternative methodological approach to project AMUQ by 2040. Specifically, we performed a systematic literature review to identify global average AMUI parameters for key livestock species, including cattle, pigs, chickens, sheep, and aquaculture. These parameters were drawn from previous studies on livestock AMUQ projections. For example, AMUI parameters for pigs (173.1 mg/kg), cattle (59.6 mg/kg), chickens (35.4 mg/kg), and sheep (243.3 mg/kg) were obtained from Mulchandani et al.[4]. For aquaculture, an average AMUI parameter of 208 mg/kg was used, obtained from Schar et al.[17] which aggregated AMUI data from 12 diverse countries representing different aquaculture species. To generate alternative projections of AMUQ by 2040, we employed Eq. (15), where AMUQ represents the total global quantity of antibiotic for a species $s$ at time $t$, LBIO is the expected quantity of livestock biomass for species $s$ at time t, and AMUI is the global average antibiotic use intensity in mg/kg of biomass for species $s$ at time $t$.

$$AMUQ_{st} = \sum_{i=1}^{n} LBIO_{st} \times AMUI_{st} \qquad (15)$$

### Reporting summary

Further information on research design is available in the Nature Portfolio Reporting Summary linked to this article.

## Data availability

The data utilized in this study were obtained from various sources and institutional databases. While some datasets are publicly accessible, others contain confidential information and are subject to restricted access. Below is a detailed list of the data sources and their access information. FAOSTAT Dataset: The livestock population data for terrestrial animals were sourced from the FAOSTAT database. These data are publicly available and can be accessed at https://www.fao.org/faostat/. FishStatJ Dataset: Data on aquaculture species were obtained from the FAO FishStatJ database. These publicly available data can be accessed at https://www.fao.org/fishery/en/statistics/software/fishstatj World Population Prospects Dataset: The human population growth projections under medium fertility variant were sourced from the United Nations' Population Division Data Portal. These projections are publicly available at https://population.un.org/dataportal/home?df=1ad6ba13-4c12-49be-8955-1582c64990bf. Global Livestock Environmental Assessment Model (GLEAM) Dataset: Herd structure data were obtained from FAO's GLEAM. Data use is subject to FAO's confidentiality agreements, which may restrict sharing and usage. Researchers interested in accessing the GLEAM dataset may contact the FAO GLEAM team at Info-GLEAM@fao.org. Requests are typically addressed within 30 days. Gridded Livestock of the World (GLW) Dataset: Spatial distribution data of livestock populations were sourced from the GLW dataset. The raw data used in this study are subject to FAO's confidentiality agreements, which may impose restrictions on sharing and usage. Researchers interested in accessing the GLW dataset may contact the FAO GLW team at GLW@fao.org. Requests are typically addressed within 30 days. Publicly available data can be accessed at https://www.fao.org/livestock-systems/global-distributions/en/. ANIMUSE—World Organization for Animal Health (WOAH) Dataset: Data on antimicrobial use (AMU) quantity were obtained from WOAH's seventh annual report on antimicrobial agents intended for use in animals. The study utilizes data aggregated at the regional level. These publicly available data can be accessed at https://amu.woah.org/amu-system-portal/home. FDA Dataset: Antimicrobial sales and distribution data in the United States for 2019 were obtained from the U.S. Food and Drug Administration (FDA). These publicly available data can be accessed at https://www.fda.gov/animal-veterinary/antimicrobial-resistance/fda-reports-data-dashboards-veterinary-antimicrobial-drug-sales-use-and-resistance. CIPARS Dataset: Antimicrobial sales and distribution data in Canada for 2019 were obtained from the Canadian Integrated Program for Antimicrobial Resistance Surveillance (CIPARS). These publicly available data can be accessed at https://health-infobase.canada.ca/veterinary-antimicrobial-sales/ Source data are provided with this paper.

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

## Acknowledgements

This research was conducted by the Livestock Policy Lab (LPL), a science-policy platform hosted by the Animal Production and Health Division at the Food and Agriculture Organization of the United Nations (FAO). We extend our deepest gratitude to FAO, particularly the Animal Production and Health Division and the Fisheries and Aquaculture Division, for the invaluable contributions of its staff and for facilitating access to essential datasets. Our sincere appreciation also goes to the Fleming Fund project for providing the financial support that made this research possible. We wish to acknowledge the World Organization for Animal Health (WOAH), particularly the team of experts from the Antimicrobial Resistance and Veterinary Products Department, for their technical feedback on the preliminary version of the manuscript. Additionally, we are grateful to the Quadripartite AMR Economic Working Group for the engaging discussions and valuable insights that informed this work. The views expressed in this publication are those of the authors and do not necessarily reflect the views of the Food and Agriculture Organization of the United Nations (FAO). Any errors, omissions, or misinterpretations remain the sole responsibility of the authors.

## Author contributions

A.A.: Conceptualization, Methodology, Formal analysis, Writing—Original Draft, Review and Editing, Supervision. W.T.: Formal analysis, Data Curation, Writing—Review and Editing, Investigation. F.N.: Formal analysis, Writing—Review and Editing. T.B.: Formal analysis, Writing—Review and Editing. G.C.: Visualization. J.S.: Resources, Writing—Review and Editing, Funding acquisition, Supervision.

## Competing interests

The authors declare no competing interests.
