## [Transparent Peer Review file · Nature Communications]

The Future of Antibiotic Use in Livestock

Corresponding Author: Dr Alejandro Acosta

Version 0:

Reviewer comments:

Reviewer #1

(Remarks to the Author)

Review of "Antimicrobial Use in Livestock: Future Global Perspectives"

This is an interesting paper on an important topic. The authors calculate projections of AMU globally and across different regions in the world. Change in AMU is strongly driven by changes in animal production required to produce sufficient protein for (growing) human populations. In addition, changes in use patterns are expected in some regions (in particular in (Northern) Europe, because of changing policies or regulations. The authors discuss in their closing paragraph in the discussion some of the major limitations of the study: availability of public disaggregated data and the extent of data coverage regarding AMU in animals are considered important issues, in addition to lack of differentiation for animal species. This shows that the authors are aware of limitations of this exercise. They could give more information how the data limitations affect their analyses and outcomes. However, the paper is relatively technical and more emphasis should be given to the narrative of the paper and in particular the transparency regarding the description of the methodology. Although a supplement is included, the core paper should give more details about the approach chosen and underlying assumptions. For instance, regional AMU is difficult to compare between continents or regions (fig 4 and 5) because AMU is expressed in tons. No mention is made of AMU per unit of weight of the animals. So, are the differences the result of differences in animal weight, use patterns or both, what are driving factors per region? Possibly, adjustments have been made by unit of weight, but this is not mentioned in the legend of the figures and this information should be given to support an appropriate interpretation of the paper.

The paper could be strengthened by including more explicitly sensitivity analyses in which some of the assumptions are more critically explored and changed. For instance, now, AMU reduction scenario's of -30% are being explored, but data from Denmark, The Netherlands and Belgium over the last years show that larger reductions seem possible, with reductions up to 70% in different animal species. These policies will find wider introduction because of new EU regulations making monitoring of AMU mandatory the coming years, possibly leading to further reductions in AMU.

Specific comments

Line Comment

29 There are also studies suggesting that the impact of AMU in livestock has a more limited impact on AMR in humans. Also see: [https://doi.org/10.1016/S2542-5196\(19\)30130-5](https://doi.org/10.1016/S2542-5196(19)30130-5). I suggest to nuance the statement made here.

30 The references "Woolhouse et al., 2016" and "Gameda et al., 2020" do not support the statement made here. The association between AMU in livestock and AMR in humans was not assessed in these studies.

35 PCU does not reflect average weight correctly because the time of exposure to antimicrobials is shorter than a year for species with a shorter lifespan. As a result, the PCU overestimates animal weight for these species. See also: Comparing human and animal antimicrobial usage: a critical appraisal of the indicators used is needed - PubMed (nih.gov)

Line 38 A reference is made to Radke, 2017, in this study a PCU adjusted by lifespan is suggested to overcome the issue of not correcting for lifespan in the PCU for short-lived species. The following paragraph suggests that in this study a correction is made for lifespan, but I believe only a correction is in the used weights to calculate the PCU is made.

Line 51 Intensity. The word 'Dose' is to be preferred here.

Line 53 The PCU method is not mentioned in this paragraph. I suggest to mention it here and make a brief comparison to the LBC method to show the main differences. It seems to make sense that the PCU method

Line 73 Biomass has multiple interpretations in literature. "Livestock biomass" is not explained here, please describe how it is interpreted here under the used LBC method. Also not that PCU leads to lower AMU results than the LBC method (line 101) which is clearly described in the reference mentioned earlier (see comment line 35).

Line 87 Please provide information on the data used to determine the baseline AMU. It is unclear what is the basis of this number, how many countries are in the WOA report for example? What is the coverage of these data? How (un)certain is the baseline AMU. This is very important as this level is used as a basis for the projections of AMU by 2040.

Line 111 Sensitivity analysis is required here with regard to the Swine fever effect. What would resulting AMU be if no swine fever would have occurred?

Line 172 How are these upper and lower bound projections made for livestock biomass?

Line 174 Why was a 30% decrease chosen, please elaborate on this specific number.

Line 194 Please explain the figure here very briefly to make the figure easier to interpret.

Line 226 Please provide some insight in the differences stated here.

Line 240 The accuracy of this projection (and the other projections) also highly depends on the accuracy of the baseline estimate for AMU, this could be more clearly stated in the discussion.

Line 451 Suggest to put this in the main text. Also it would be interesting to specify more which countries are missing.

Line 565 Why $1 + n_{i,j}$? So, if an animal species has 2 production cycles you multiply the average number of animals by 3 for the PCU calculation?

Line 628 How is AMU intensity calculated for regions where AMU is unknown?

Reviewer #2

(Remarks to the Author)

NCOMMS-24-36673

Antimicrobial Use in Livestock: Future Global Perspectives

This manuscript seeks to explore trends in the quantity of antimicrobial use (AMU) in livestock due to concerns about its role in antimicrobial resistance (AMR). The authors do this with large, decades-long datasets and predictive methodologies. The goal was to improve the existing research using more recent biomass data, improve the estimation of average animal weight that accounts for animal production cycle length, predict AMU, and explore future AMU trajectories. The authors attempted to predict AMU for different geographic regions based on projected growth in the number of animals of different species and corresponding production categories (i.e., their animal biomass), which are tied to the increasing human population (and the need for food). They project an increase to 138,833 tons of AMU by the year 2040, but include scenario analysis to account for potential future changes in the livestock and AMU arena. The study's main finding is that the AMU in livestock may continue to grow due to the projected increase in livestock biomass (to accommodate the needs of the growing human population). This reviewer has several concerns regarding the seemingly simplistic methodology, incomplete explanation of the methodology, and superficial interpretation of findings.

Main comments:

1. The study's finding that the increase in livestock biomass will be accompanied by an increase in AMU is intuitive and could be derived through a back-of-the-envelope calculation. The main contribution of this study, however, is the provided numerical predictions of how bad (or not) the AMU may be in 2040 globally and in different regions and subregions. However, the key ingredient in these numerical predictions is AMU sales data, which are known to be inaccurate and are just an 'easy-to-obtain' proxy of the actual antimicrobial use; the sales data provide no insights about AMU reasons in a country and can't explain the nuances in the characteristics of the animal population exposed to antimicrobials (e.g., age and weight at treatment), their health status, availability of drugs and their potency, nor the accuracy or completeness of sales records. Thus, while I support the need for AMU projections, basing those projections on the region and subregion-specific AMU sales data may be misleading for two reasons: (1) biases affecting the local AMU sales - livestock biomass relationship are unknown and unaccounted for; and (2) the authors emphasize the increasing AMU (based on sales data), while the underlying reason for increasing AMU reason in terms of increasing animal biomass is ignored. Interpretation of study results should account for these nuances to avoid misinterpretation.
2. For transparency, the authors should provide a figure of the projected livestock biomass using PCU and LBC methods so that the reader can interpret the projected increase in AMU (whether it is proportional to livestock biomass or not). I suggest a figure showing both the projected biomass and AMU.
3. Why was aquaculture omitted from the biomass estimation and AMU prediction? Does the AMU sales data used in the calculation include the sales for use in aquaculture? If AMU sales data include the sales for use in aquaculture but biomass was not calculated for aquaculture, does that bias predictions? In which direction? And how does that differ for different (sub)regions in the world?
4. What assumptions about environmental, political, or socioeconomic state of the world must be made in order for the BAU AMU trends to be considered accurate predictions of the year 2040?
5. Were there available historic data for calculating AMU intensity prior to the year 2019: so that trends in AMU intensity could be projected similarly to the projections of livestock biomass?
6. In the scenario analysis, is there a reason why a scenario was not tested in which livestock biomass remained at the lower bound, while AMU remained at BAU (similar to S1)?
7. A few questions concerning the original GLW4 dataset would benefit from discussion: how does the inclusion of equines, which while considered livestock are not frequently raised as such, contribute to the projections of AMU? Similarly, while this is a small fraction of the food animal population, what about livestock animals raised for companion purposes (a growing trend)? The focus on AMU in livestock is typically understood to mean AMU in animals raised for food/fiber/draft, and not necessarily AMU in large animal species as a whole. Finally, does this dataset include drugs like coccidiostats that are technically antimicrobials but fed to ruminants for greenhouse gas mitigation purposes?

Specific Comments

8. Lines 12-13: I would amend this statement to be more neutral, as the rapid increase in AMU absolute quantities itself does not necessarily pose a threat to public health; more, it is misuse and poor stewardship of such large quantities.
9. Lines 23-25: As the statement mentions disease in both humans and animals, please provide a citation that enumerates the number of animal deaths attributed to AMR.
10. Lines 28-30: The statement "Notably, using antimicrobials in livestock production has been identified as a key driver of AMR in both humans and animals (Ardakani et al., 2023; Woolhouse et al., 2016; Gemeda et al., 2020)." is not supported by the cited studies. Ardakani reports correlations (not causative relationship) between AMU in livestock and AMR in humans. Woolhouse is a review study, and Gemeda is a cross-sectional interview-based study of human perceptions/knowledge. Thus, none of these citations substantiate the wording that AMU in livestock is a 'key driver' of AMR in animals and humans. While the causal relationship is suspected, clear evidence of a causal relationship (at least on a scale that would justify the wording "key driver") is still lacking. Notably, sales AMU data will NOT provide evidence of a causal relationship, and advocating for the importance of such data (line 21) is missing the point. Please adjust Lines 28-30 to reflect the state of knowledge about the causative relationship between AMU in livestock and AMR (or provide references that support such causal relationship). In line 21 (and elsewhere, e.g., lines 217-218), I would rather see that the authors advocate for accurate animal-level data on AMU in livestock (rather than AMU sales data) because, unlike AMU sales data, surveillance of animal-level data will allow causal inference between AMU and AMR.
11. Figure 1, step 3: The world human population dataset was used as an exogenous variable to project the livestock population growth. How exactly was that done?
12. Lines 66-68: Please list the species included in this section rather than in the Supplementary Material.
13. Lines 87-89: Does this dataset include AMU both on- and extra-label?
14. Lines 100-101: The original PCU by the ESVAC of the EMA in the EU was developed to represent the animal's average weight at treatment in EU countries. It is unclear if, in this study, the authors used the original PCU methodology (developed and parametrized for the member countries of the EU) or if the authors have modified PCU to reflect animal weight at slaughter in different countries (see equations 7 and 8 in supplemental materials). The authors failed to explain why PCU consistently predicts less AMU than LBC. If PCU measures the weight at treatment while LBC reflects the weight at slaughter, then, in fact, PCU is a more correct reflection of the actual AMU use, albeit it requires knowledge of the weight at treatment for the livestock population in different (sub)regions. The differences in derivation and the estimated values for PCU and LBC must be explained. This is important because of considerable differences in estimates in some of the world's regions, and the reason for such differences needs to be explained.
15. Lines 110-111: Did this sharp decline in AMU due to decreases in swine biomass from ASF have an appreciable impact on your projections for this region?
16. Lines 113-116: how is the 95% CI estimated? What does the 95% CI reflect? Is that the uncertainty in the projected livestock population growth or something else?
17. Lines 136-140: please comment on whether this high proportion (73%) comes from the projected livestock growth (and/or projected human population growth), and/or projected increases in AMU.
18. Lines 140-141: "This concentration underscores the pivotal role these subregions play in the global AMU landscape within the livestock sector." Related to the previous comment (Main Comment #1), it seems misleading to say that these subregions will play a pivotal role in AMU in the livestock sector without acknowledging that the projected AMU growth reflects the projected animal biomass growth.
19. Lines 241-243: This is an overly simplistic view that ignores other factors that have nothing to do with AMU, and which can be expected to affect the trajectories, including human consumption preferences (different livestock species and production categories have different needs for AMU), livestock numbers, livestock health, and the availability of and access to methods to prevent infections in different regions.
20. Lines 244-253: The main unrecognized limitation is the quality (and potential bias in an unknown direction) of the used AMU sales data.
21. Line 249-251: See above Main Comment #7 related to the inclusion of species and production settings for this aggregated dataset.
22. Lines 648-649: "Our analysis encompasses primary livestock species, such as cattle, pigs, chicken, goats, and sheep." Please specify exactly which species were considered in this analysis. Based on this and previous text, for example, it is unclear if AMU in horses is included in the analysis.
23. Lines 662-663: Does the "variability in livestock biomass" account for increases in mean bodyweight due to genetic selection for larger animals?
24. Minor comment: mass and weight are used interchangeably, though mass would seem to be a more appropriate term.

Reviewer #3

(Remarks to the Author)

Reviewer #4

(Remarks to the Author)

The authors investigate an important area of antimicrobial resistance, the use of antibiotics in the livestock sector. The analysis is at a global scale and therefore has to make some compromises in terms of disaggregation of detail in terms of biology, geography and time. These limitations are not dealt with in the discussion or introduction and I would have expected a need for:

- antimicrobials used - the focus of the paper is antibiotics, not all antimicrobials
- antibiotics are not homogenous have different importances in terms of their use in livestock and people, different half lives in the environment and their use generates different resistant profiles
- there remains uncertainty of how important livestock derived AMR is to human health
- there also remains uncertainty of the impacts on livestock production efficiency with reduction of antibiotic use in livestock, this has implications of efficiency of land and water use and hence possible impacts of the environment as well as impacts on consumer welfare
- country level data is where we would begin to have some potential interventions questions that can be dealt with by jurisdiction, understandably the authors have not gone to country level, yet this is where power lies for change

With regards the temporal issues the analysis presented is dependent on two parameters, the biomass of the animals and the level of antibiotic use per biomass unit. These parameters have a high degree of uncertainty:

- reliance on FAOSTAT data for populations and animal weights is fraught with difficulties when estimating biomass and the methods to estimate biomass are not decided upon
- antibiotic use per unit of biomass is not well collected and published even in high income countries

I would recommend that more time is spent on these elements in the discussion as the paper is a guide to what has happened and what might happen, whereas it probably is as useful in terms of setting out what data needs to be collected or what data collection needs to be improved to reduce the uncertainty of current estimates and to improve the predictions of the estimates generated.

I would also recommend the authors look at the following paper:

Schroback, P., Dennis, G., Li, Y., Mayberry, D., Shaw, A., Knight-Jones, T., Marsh, T.L., Pendell, D.L., Torgerson, P.R., Gilbert, W., Huntington, B., Raymond, K., Stacey, D.A., Bernardo, T., Bruce, M., McIntyre, K.M., Rushton, J., Herrero, M., 2023. Approximating the global economic (market) value of farmed animals. *Global Food Security* 39, 100722. <https://doi.org/10.1016/j.gfs.2023.100722>

And if possible access the short studies that were carried out by OECD on antibiotic use in China and Brazil approximately 5 years ago. These had very large variations from the published data at the time.

Version 1:

Reviewer comments:

Reviewer #2

(Remarks to the Author)

NCOMMS-24-36673A

Antimicrobial Use in Livestock: Future Global Perspectives

The authors have adjusted the scope of the manuscript, including shifts in verbiage that do not change the fundamental conclusions of the work, but refine the presentation in a way that is a significant improvement upon the previous iteration. As co-reviewers, we are pleased with the increased quality of this work and recommend it for publication. We have only one minor remaining comment:

Line 453-454: What is meant by (and what are the implications of) "coordinated effort that targets reductions in LBIO?" This might be misinterpreted as "the solution to rising livestock AMUQ is to cull the animals" to reduce the animal biomass. With growing human populations worldwide, the requirements for animal-derived proteins will likely increase. While studies that "optimize" the human diet in terms of global cropping capabilities (e.g. EAT-Lancet Report, 2019) suggest a decreased reliance on animal proteins, this is not as feasible in developing countries where LBIO is intimately tied to the well-being of people living below the poverty line. Is it appropriate or feasible to highlight this LBIO-reduction intervention, even if it is mathematically sound? To avoid misinterpretation, we recommend revising the statement or adding a disclaimer to clarify.

Reviewer #3

(Remarks to the Author)

Reviewer #4

(Remarks to the Author)

General comment

- use of acronyms does not help the reader and some of the acronyms used are not common and in some cases not explained, for example LBIO, AMUI - do a search a replace and get rid of them

Given you have decided to develop a different means of calculating biomass you should look at the GBADs technical guide (<https://animalhealthmetrics.org/gbads-technical-guide/>) which has a section on biomass estimations. And there should be a comparison between the biomass from the PCU and the LBC methods to show the differences.

Figure 2 does not seem to make sense. The AMU per unit of biomass (which I think you refer to as AMUI) should be an input variable to the estimation of the quantity of AMU overall (which I think is referred to as AMUQ) not something that changes with the way biomass is calculated. Have I missed something in the calculations or is there something that needs more explanation? There is a circularity of the estimation of the AMU per unit of biomass that does not feel very sound. Where is the use of reported AMU overall coming into the calculations and has there been any attempt to relate the biomass in different production systems as well as different species. For example dairy versus beef in cattle or broiler versus layers in chickens. These then begin to unravel some of the critical questions on data

Line 525-26 - states that the biomass calculation used is more accurate than the PCU, what has been the gold standard to allow this statement to be made?

- Given that you have estimated biomass by species, production systems, age and sex it would make sense to present an estimate of the AMU broken down more carefully. If you cannot do this by age and sex then at least by production system. This is particularly important in the cattle and pigs which have the highest estimated use of antibiotics

- No mention or discussion of the type of antibiotic and this is critical when thinking through the implications and the authors should be encouraged to look at the recent paper by Narmada et al (2024) <https://doi.org/10.1186/s12879-024-09847-3>

Specific

- line 36 "diving"?

- line 47 First spelt incorrectly

- line 244 - do you mean "developing countries"? should this not read low and middle income countries, or indeed should it not read high income countries. There is little evidence that livestock numbers are reducing in the LMICs, it is the reverse. The downward trend in high income countries have been a reduction in animal numbers but an increase in average animal size in the case of cattle and a faster turnover in the case of poultry and pigs

Version 2:

Reviewer comments:

Reviewer #4

(Remarks to the Author)

No further observations on the paper, I recommend it is published

Antibiotic Use in Livestock: Future Global Perspectives

Alejandro Acosta, Wondmagegn Tirkaso, Francesco Nicolli, Thomas P. Van Boeckel, Junxia Song

NCOMMS-24-36673

Reply to reviewer's comments

October 15, 2024

Contents

Reviewer 1	2
Reviewer 1 specific line comments	3
Reviewer 2	8
Reviewer 2 specific line comments	11
Reviewer 3	17
Reviewer 4	17

Reviewer 1

Thank you for taking the time and effort to review our work. We are truly grateful for your comments. As you will notice, we have carefully addressed each one to make the most of your feedback.

Comment 2. Give more information on how the data limitations affect their analyses and outcomes.

- Reply: Thank you for your comment. We have significantly expanded the discussion in the limitations section, providing a more detailed analysis of the data limitations and their potential impact on our findings. Specifically, we have included an in-depth examination of how the current AMU data may introduce potential biases into the AMU projections, particularly due to gaps in reporting and variations in data accuracy. **(Please see page 21, line 477)**

Comment 3. The paper is relatively technical, and more emphasis should be given to the narrative of the paper and in particular the transparency regarding the description of the methodology.

- Reply: Thank you for your comment. We have made significant improvements to the narrative of the paper to ensure it is more accessible to non-technical experts. We have also expanded and clarified the description of the methodology, providing additional context and step-by-step explanations. **(Please see page 3)**

Comment 4. Although a supplement is included, the core paper should give more details about the approach chosen and underlying assumptions.

- Reply: Thank you for your comment. As requested, we have revised the methodological part of the paper including a detailed description of the different approaches employed, ensuring that key methodological aspects are clearly described and explained. Additionally, we have provided comprehensive information on the main underlying assumptions that guide our methodology. **(Please see page 3)**

Comment 5. For instance, regional AMU is difficult to compare between continents or regions (fig 4 and 5) because AMU is expressed in tons. No mention is made of AMU per unit of weight of the animals.

- Reply: Thank you for your comment. The objective of Figure 4 (previously Figures 4 and 5) is to illustrate the absolute magnitude of AMUQ by region, offering an overview of regional contributions to global AMUQ. Please note that we report AMUI per regions in figure 2b. As highlighted, we assume AMUI remains constant over time in the business-as-usual scenario, hence the focus in Figure 4 remains on projecting total AMUQ across regions. We have ensured that the distinction between AMUQ and AMUI is clearly highlighted in the paper to aid in understanding both absolute and relative usage patterns. **(Please see page 12)**

Comment 6. So, are the differences the result of differences in animal weight, use patterns or both, what are driving factors per region?

- Reply: Thank you for your comment. Thank you for your comment. The differences in the AMUQ projections arise from a combination of changes in livestock biomass and antimicrobial use intensity (AMUI) patterns under a BAU scenario, both of which vary by region. Key drivers of livestock biomass differences across regions include population growth and income growth,

which influence livestock production systems. The variations in AMUI are simulated by considering reductions of 30% and 50%, in alignment with the targets set by the Muscat Declaration.

- **Comment 7.** Possibly, adjustments have been made by unit of weight, but this is not mentioned in the legend of the figures and this information should be given to support an appropriate interpretation of the paper.
- **Reply:** Thank you for your feedback. We would like to clarify that the majority of the AMUQ results reported are expressed in tons using the LBC approach. We would be more than happy to revise the specific figures you are referring to.
- **Comment 8.** The paper could be strengthened by including more explicitly sensitivity analyses in which some of the assumptions are more critically explored and changed. For instance, now, AMU reduction scenario's of -30% are being explored, but data from Denmark, The Netherlands and Belgium over the last years show that larger reductions seem possible, with reductions up to 70% in different animal species. These policies will find wider introduction because of new EU regulations making monitoring of AMU mandatory the coming years, possibly leading to further reductions in AMU.
- **Reply:** Thank you for your comment. We acknowledge that larger reductions have been achieved in certain regions, particularly in Europe. However, since our study aims to use a global target scenario, we have accounted for varying rates of progress across different regions. Nevertheless, and following the AMUI reduction targets of 30% and 50% set in the Muscat Manifesto, adopted at the Global High-Level Ministerial Conference on AMR in Oman in 2022, we have taken your suggestion into consideration and integrated an even more optimistic scenario reflecting a 50% reduction in AMUI by 2040. This inclusion allows our study to explore the potential outcomes under more ambitious assumptions. **(Please see pages 15-17)**

Reviewer 1 specific line comments

Comment 9. Line 29. There are also studies suggesting that the impact of AMU in livestock has a more limited impact on AMR in humans. Also see: [https://doi.org/10.1016/S2542-5196\(19\)30130-5](https://doi.org/10.1016/S2542-5196(19)30130-5). I suggest to nuance the statement made here. **Line 30.** The references “Woolhouse et al., 2016” and “Gemedda et al., 2020” do not support the statement made here. The association between AMU in livestock and AMR in humans was not assessed in these studies.

- **Reply:** Thank you for these comments. We have decided to frame the motivation of the paper around the recent commitments made by governments to reduce AMU quantity, during the third high-level ministerial conference on AMR, held in Muscat in 2022, governments committed to reducing the total amount of AMU in agrifood systems globally and the United Nations General Assembly (UNGA) high-level meeting on AMR in September 2024. We believe this new motivation is better aligned with the purpose and findings of the paper. Additionally, we have integrated some of these comments into the discussion section. **(Please see page 2)**

Comment 10. Line 35. PCU does not reflect average weight correctly because the time of exposure to antimicrobials is shorter than a year for species with a shorter lifespan. As a result, the PCU overestimates animal weight for these species. See also: Comparing human and animal antimicrobial usage: a critical appraisal of the indicators used is needed - PubMed (nih.gov)

- Reply: Thank you for this comment. The comment you provided aligns with our argument regarding the potential biases in AMU estimates when using PCU. We appreciate you bringing this paper to our attention and have now included it as a reference in our study. **(Please see line 39)**

Comment 11. Line 38. A reference is made to Radke, 2017, in this study a PCU adjusted by lifespan is suggested to overcome the issue of not correcting for lifespan in the PCU for short-lived species. The following paragraph suggests that in this study a correction is made for lifespan, but I believe only a correction is in the used weights to calculate the PCU is made.

- Reply: Thank you for this comment. Indeed, following Radke (2017), we have considered animal lifespan in the calculation of livestock biomass, as captured by the Livestock Biomass Consumption (LBC) metric. Specifically, as outlined in equations ix and x of the methodology, the parameter n_i denotes the number of production cycles used to account for lifespan differences among species (i). We have clarified this point in the methodology to avoid any confusion regarding how lifespan is addressed in our study. **(Please see line 154)**

Comment 12. Line 51. Intensity. The word 'Dose' is to be preferred here.

- Thank you for your suggestion. We understand your suggestion to use the term "dose" but we prefer to retain "intensity" as it better captures the concept we aim to convey. "Intensity" refers to the relative amount of AMU per unit of livestock biomass, which aligns with the focus of our analysis on the broader patterns of AMU. In response to your comment, we have clarified how we define "intensity" within the context of our study in the methodology.

Comment 13. Line 53. The PCU method is not mentioned in this paragraph. I suggest to mention it here and make a brief comparison to the LBC method to show the main differences. It seems to make sense that the PCU method.

- Reply: Thank you for this comment. We have revised the methodological section and include a detailed description of the PCU method, emphasizing the key differences between it and the LBC method. In particular, we describe in detail how average livestock biomass is calculated in both approaches. **(Please see line 136)**

➤

Comment 14. Line 73. Biomass has multiple interpretations in literature. "Livestock biomass" is not explained here, please describe how it is interpreted here under the used LBC method. Also not that PCU leads to lower AMU results than the LBC method (line 101) which is clearly described in the reference mentioned earlier (see comment line 35)

- Reply. Thank you for your comment. We have clarified the definition of livestock biomass in the context of the LBC method (**Please see line 130**). Regarding your note on the PCU leading to lower AMU results compared to the LBC method, this is indeed one of the key arguments and findings of our paper. We contend that the use of PCU could result in an underestimation of AMU quantities in livestock. This discrepancy is expected, as the PCU method often estimates lower levels of biomass by using average weight at slaughter age as a proxy for live weight, while the LBC method utilizes more granular data on live weight by commodity, production system, at cohort level. (**Please see figure A1 in Annex 1**)

Comment 15. Line 87. Please provide information on the data used to determine the baseline AMU. It is unclear what is the basis of this number, how many countries are in the WOAHA report for example? What is the coverage of these data? How (un)certain is the baseline AMU. This is very important as this level is used as a basis for the projections of AMU by 2040.

- Reply: Thank you for your comment. We have clarified the characteristics of the WOAHA data used to establish the baseline AMU (see Section 2.5). Additionally, we have expanded the discussion section to provide a detailed analysis of the uncertainties and limitations associated with this dataset. We utilized WOAHA's seventh annual report on antimicrobial agents intended for use in animals. The data was collected through surveys from 157 countries, with 121 providing quantitative information, representing approximately 85% of WOAHA members and 70% of the global animal biomass. (**Please see line 222**)

Comment 15. Line 111 Sensitivity analysis is required here with regard to the Swine fever effect. What would resulting AMU be if no swine fever would have occurred?

Reply: Thank you for your comment. This observation was also raised by another reviewer, underscoring its relevance. Assessing the impact of a potential structural break in the forecasted data generation process is indeed a compelling area for further research. However, it is important to note that the time series returns to its long-term equilibrium relatively fast, indicating that the shock may not have had a lasting effect on the AMUQ projections. This implies that the shock may not have had a lasting effect on AMUQ projections, demonstrating the model's ability to account for the speed of adjustment and correct for deviations from the long-run trend. This is an area of research of particular interest to us, as we have conducted similar studies analyzing the effects of animal disease outbreaks on market dynamics, such as African Swine Fever (ASF) in China and Highly Pathogenic Avian Influenza (HPAI) in Mexico. These studies may serve as useful references for further exploration of this topic:

- Acosta, A., Lloyd, T., McCorriston, S., & Lan, H. (2023). The ripple effect of animal disease outbreaks on food systems: The case of African Swine Fever on the Chinese pork market. *Preventive Veterinary Medicine*, 215, 105912.
- Acosta, A., Barrantes, C., & Ihle, R. (2020). Animal disease outbreaks and food market price dynamics: Evidence from regime-dependent modelling and connected scatterplots. *Australian Journal of Agricultural and Resource Economics*, 64(3), 960-976.

We have noted this as an important area for future research.

Comment 16. Line 172. How are these upper and lower bound projections made for livestock biomass?

- Reply: Thank you for your comment. The upper and lower bound projections scenarios for livestock biomass were calculated using the confidence intervals derived from our statistical models, specifically reflecting the uncertainty in the regression coefficients. In this way we simulate the projected upper and lower changes in livestock population if a 95% confidence interval is considered. This method allows us to partially capture the degree of uncertainty in our projections providing a range of plausible outcomes rather than a single estimate. We have clarified our approach in the methodological section.

Comment 17. Line 174. Why was a 30% decrease chosen, please elaborate on this specific number.

- Thank you for your comment. The 30% decrease in AMU intensity was selected based on the targets established in the Muscat Manifesto, adopted during the Global High-Level Ministerial Conference on Antimicrobial Resistance in Oman in 2022. The Muscat Manifesto calls for significant reductions in antimicrobial use in food-producing animals to combat the growing threat of antimicrobial resistance (AMR). Further, in response to your suggestion for a more ambitious target, we have also included a scenario that models a 50% reduction in AMU intensity, as also indicated in the declarations of the Muscat Manifesto. This clarification is now highlighted in the methodological section of the manuscript. Please note that these targets were dropped from the UNGA declaration.

Comment 18. Line 194. Please explain the figure here very briefly to make the figure easier to interpret.

- Reply. Thank you for your comment. To enhance clarity, a brief explanation has been added to each figure. Figures 6 and 7 illustrate the projected livestock AMUQ across the eight scenarios outlined in Table 1, which combine different levels of LBIO and AMUI. These scenarios range from a Business-as-Usual (BAU) trajectory, where both LBIO and AMUI remain constant, to various combinations of upper and lower LBIO bounds with AMUI reductions of 30% or 50%. This added context should make it easier to interpret the figures and understand the potential impact of different policy and management interventions on AMU projections.

Comment 19. Line 226. Please provide some insight in the differences stated here.

- Reply. Thank you for your comments. We have expanded the discussion to clarify the methodological differences between Mulchandani's approach and ours. One key distinction is that Mulchandani's AMU projections are based on parameters from the FAO report "The Future of Food and Agriculture to 2050" (FOFA). Their approach uses FAOSTAT animal numbers for 2020 and adjusts them by the ratio of projected animal stocks under FOFA's BAU scenario for 2020-2030. It's worth noting that FOFA uses 2012 as the baseline year and employs a global partial equilibrium model for its projections. In contrast, our study utilizes time series data from 1961 to 2021 and applies an ARDL forecast model, offering updated projections for 2020-2040. Additionally, there are significant differences in how AMU intensity is calculated. Mulchandani's

study uses the PCU approach, which estimates animal biomass based on slaughter weight. Our analysis, however, adopts the LBC method. This approach considers a more detailed livestock herd structure and accounts for live weight by commodity, production system, and cohort, providing a more nuanced and granular assessment of AMU intensity compared to the PCU method. (Please see line 426)

Comment 20. Line 240. The accuracy of this projection (and the other projections) also highly depends on the accuracy of the baseline estimate for AMU, this could be more clearly stated in the discussion.

- Reply: Thank you for your comment. We fully agree that the accuracy of the projections is directly linked to the precision of the baseline AMU estimates. In the revised discussion, we have clarified this point. Additionally, we have expanded on how several factors—such as global and regional policies, economic incentives, subsidies, and environmental conditions—will play a significant role in shaping these projections. **(Please see line 466)**

Comment 21. Line 451. Suggest to put this in the main text. Also it would be interesting to specify more which countries are missing.

- Thank you for your suggestion. The text has now been incorporated into the methodological sections. While we agree that providing country-specific AMU information would offer valuable insights, the data confidentiality agreement prevents us from listing the reporting and non-reporting countries.

Comment 22. Line 565. Why $1 + n_{i,j}$? So, if an animal species has 2 production cycles you multiply the average number of animals by 3 for the PCU calculation?

- Thank you for your comment. That's an excellent point. Please note that this equation is used in both Tiseo (2020) and Mulchandani (2023) papers. However, I fully agree with you—I don't see the reason to add 1 in the equation. That said, please note that we are using the LBC, not the PCU. In the LBC, we multiply the total number of animals directly by n . **(Please see line 148)**

Comment 23, Line 628. How is AMU intensity calculated for regions where AMU is unknown?

- Thank you for your comment. For regions where AMUI data is not available, we have applied an extrapolation method to account for non-reporting countries. This involves the following steps:
 1. Estimation of LBIO for Reporting Countries: We begin by estimating the LBIO for the group of reporting countries within a region.
 2. Calculation of AMUI for Reporting Countries: The AMUI for reporting countries is then calculated by dividing the AMUQ for these countries by their corresponding LBIO, providing an intensity measure of AMU per unit of livestock biomass.

3. Extrapolation of AMUI to Non-Reporting Countries: Finally, we assume that non-reporting countries have a similar AMUI to the reporting countries. We estimate the total AMUI for the entire region by applying the calculated AMUI to the total livestock biomass of both reporting and non-reporting countries.

This methodology has been clarified in response to your comment and is now fully detailed in Section 2 of the paper. **(Please see line 173)**

Reviewer 2

Thank you for taking the time and effort to review our work. We are truly grateful for your comments. As you will notice, we have carefully addressed each one to make the most of your feedback.

Comment 24. The study's finding that the increase in livestock biomass will be accompanied by an increase in AMU is intuitive. The main contribution of this study, however, is the provided numerical predictions of how bad (or not) the AMU may be in 2040 globally and in different regions and subregions. However, the key ingredient in these numerical predictions is AMU sales data, which are known to be inaccurate and are just an 'easy-to-obtain' proxy of the actual antimicrobial use; the sales data provide no insights about AMU reasons in a country and can't explain the nuances in the characteristics of the animal population exposed to antimicrobials (e.g., age and weight at treatment), their health status, availability of drugs and their potency, nor the accuracy or completeness of sales records. Thus, while I support the need for AMU projections, basing those projections on the region and subregion-specific AMU sales data may be misleading for two reasons: (1) biases affecting the local AMU sales - livestock biomass relationship are unknown and unaccounted for; and (2) the authors emphasize the increasing AMU (based on sales data), while the underlying reason for increasing AMU reason in terms of increasing animal biomass is ignored. Interpretation of study results should account for these nuances to avoid misinterpretation.

- Reply: Thank you for your comment. You raise important concerns about the use of AMUQ sales data as a proxy for actual AMUQ in livestock. We fully acknowledge that this proxy has limitations, particularly because AMUQ sales data may not capture critical factors such as the reasons you highlighted. Moreover, as you noted, the accuracy of sales data can vary, and potential biases affecting the relationship between AMU sales and livestock biomass are challenging to quantify.

However, please note that that without incorporating ANIMUSE sales data, the available information would be significantly limited, as only a small number of countries report actual usage data. Omitting sales data would result in overlooking a significant portion of the global context. Moreover, ANIMUSE has been internationally recognized as the reference dataset for tracking global antimicrobial usage (AMU).

Nevertheless, in response to your comment, we have revised the manuscript to explicitly address these concerns throughout the methodology, results, discussion, and limitations sections. Specifically, we now emphasize the potential uncertainties and biases inherent in relying on AMUQ sales data, highlighting that these data are often the most accessible but may not fully represent actual AMUQ practices in livestock. **(Please see line 478)**

To address the potential shortcomings of AMUQ sales data, we have conducted a robustness check using an alternative methodological approach for projecting AMUQ by 2040. This approach integrates LBIO projections with average AMUI global parameters for different animal species obtained from the literature (Mulchandani et al., 2023; Schar et al., 2020). This alternative BAU scenario allows us to assess AMUQ projections from a species-focused perspective, offering a complementary view to our global AMUQ projection estimates. **(Please see line 370)**

Comment 25. For transparency, the authors should provide a figure of the projected livestock biomass using PCU and LBC methods so that the reader can interpret the projected increase in AMU (whether it is proportional to livestock biomass or not). I suggest a figure showing both the projected biomass and AMU.

- Reply: Thank you for your comment. In response, we have added a figure to the annex that compares the projections of Livestock Biomass (LBIO) and Antimicrobial Usage Quantities (AMUQ) using the Livestock Biomass Calculation (LBC) method. However, please note that we are unable to project AMUQ using the Population Correction Unit (PCU) method, as it relies on a different livestock biomass calculation, which we have not projected in this analysis. **(Please see line 762)**

Comment 26. Why was aquaculture omitted from the biomass estimation and AMU prediction? Does the AMU sales data used in the calculation include the sales for use in aquaculture? If AMU sales data include the sales for use in aquaculture but biomass was not calculated for aquaculture, does that bias predictions? In which direction? And how does that differ for different (sub)regions in the world?

- Reply: Thank you for your comment. Initially, our projections did include aquaculture, using WOA data to estimate the proportion of AMUQ attributed to aquaculture. However, based on your suggestion, we have collaborated extensively with the FAO aquaculture team and incorporated data from FishStat to better identify the aquatic species most associated with antimicrobial use, such as salmon, catfish, shrimp, tilapia, trout, carp, barramundi, seabass, grouper, and milkfish. Additionally, we further disaggregated aquaculture data by region, which

provides a clearer understanding of how the inclusion of aquaculture influences AMU predictions. **(Please see methodological section 2, line 74).**

Comment 27. What assumptions about environmental, political, or socioeconomic state of the world must be made in order for the BAU AMU trends to be considered accurate predictions of the year 2040?

- Reply: Thank you for your comment. We have used your feedback to strengthen the narrative of the paper.

Predicting how social, economic, and environmental drivers will influence AMUQ and AMUI trajectories is complex. In our BAU scenario, we assume that socioeconomic factors like population growth and rising incomes will continue as projected, increasing demand for animal protein and driving livestock biomass growth. We also expect livestock production systems to maintain current productivity trends, with gradual improvements over time. Regarding environmental factors, the BAU scenario assumes that climate change effects will gradually intensify but will not significantly disrupt global livestock production by 2040. Livestock emissions are projected to continue along current trajectories, with only limited regulatory interventions aimed at reducing greenhouse gas emissions, primarily through reducing animal numbers in developing countries **(Please see line 244).**

However, we recognize that these assumptions may be optimistic. Climate change and policy interventions are likely to influence AMUQ pathways. For instance, stricter emissions regulations could constrain the anticipated growth in livestock biomass, while tighter antibiotic regulations could result in substantial reductions in AMUI. To account for these uncertainties, we simulate alternative scenarios for livestock AMUQ by 2040 under different assumptions (Please see Table 1). These scenarios include upper and lower bounds for LBIO projections, based on 95% confidence intervals, reflecting not only population and income growth but also the potential impact of climate conditions and environmental regulations on the livestock sector. Additionally, we simulate AMUI reduction targets of 30% and 50%, aligned with the goals of the Muscat Manifesto adopted at the 2022 Global High-Level Ministerial Conference on AMR in Oman. **(Please see methodological section 2, 253)**

- Comment 28. Were there available historic data for calculating AMU intensity prior to the year 2019: so that trends in AMU intensity could be projected similarly to the projections of livestock biomass?

- Reply: Thank you for your comment. In response we explored this aspect using available data from WOAHA, which provides a time series of AMUQ (Antimicrobial Usage Quantity) from 2015 to 2021. However, with only 8 years of data, the time series is too short to establish a robust trend for projecting AMU intensity. Additionally, the number of reporting countries providing quantitative data varied each year, making it difficult to ensure consistency and comparability across the dataset. For example, the number of reporting countries ranged from 80 in 2015 to 130

in 2019, with fluctuations in subsequent years (2020: 123, 2021: 121), which further complicates trend analysis.

Comment 29. In the scenario analysis, is there a reason why a scenario was not tested in which livestock biomass remained at the lower bound, while AMUI remained at BAU (similar to S1)?

- Reply: Thank you for your comment. We have expanded the scenario analysis to include additional scenarios, including the one you suggested. Specifically, Scenario 2 (S2) now explores a situation where livestock biomass remains at the lower bound, while AMUI follows the BAU trajectory. **(Please see line 252)**

Comment 30. A few questions concerning the original GLW4 dataset would benefit from discussion: how does the inclusion of equines, which while considered livestock are not frequently raised as such, contribute to the projections of AMU? Similarly, while this is a small fraction of the food animal population, what about livestock animals raised for companion purposes (a growing trend)? The focus on AMU in livestock is typically understood to mean AMU in animals raised for food/fiber/draft, and not necessarily AMU in large animal species as a whole. Finally, does this dataset include drugs like coccidiostats that are technically antimicrobials but fed to ruminants for greenhouse gas mitigation purposes?

- Reply: Thank you for your comment. Please note that the focus of our analysis is on animals raised for food production. The quantity of AMU in companion animals is negligible compared to that used in food-producing animals, accounting for less than 10% in high-income countries and an even smaller proportion in low- and middle-income countries. Regarding the inclusion of coccidiostats, only antibacterials with coccidiostatic properties, as referenced in the WOAHA List of Antimicrobial Agents of Veterinary Importance are included in ANIMUSE data excluding ionophores (ie: sulfonamides, arsenicals etc...).

Reviewer 2 specific line comments

Comment 31. Lines 12-13. I would amend this statement to be more neutral, as the rapid increase in AMU absolute quantities itself does not necessarily pose a threat to public health; more, it is misuse and poor stewardship of such large quantities. Lines 23-25: As the statement mentions disease in both humans and animals, please provide a citation that enumerates the number of animal deaths attributed to AMR. Lines 28-30: The statement “Notably, using antimicrobials in livestock production has been identified as a key driver of AMR in both humans and animals (Ardakani et al., 2023; Woolhouse et al., 2016; Gameda et al., 2020).” is not supported by the cited studies. Ardakani reports correlations (not causative relationship) between AMU in livestock and AMR in humans. Woolhouse is a review study, and Gameda is a cross-sectional interview-based study of human perceptions/knowledge. Thus, none of these citations substantiate the wording that AMU in livestock is a ‘key driver’ of AMR in animals and humans. While the causal relationship is

suspected, clear evidence of a causal relationship (at least on a scale that would justify the wording “key driver”) is still lacking. Please adjust Lines 28-30 to reflect the state of knowledge about the causative relationship between AMU in livestock and AMR (or provide references that support such causal relationship).

- Reply: Thank you for the comments related to lines 12-13 / 28-30. Similar concerns were raised by other reviewers, which have led us to reconsider the focus and motivation of our paper. In response, we have decided to reframe the motivation around the recent commitments made by governments to reduce AMU in agri-food systems globally. However, achieving this ambitious goal presents significant challenges, particularly in regions where livestock biomass is projected to grow. Our paper now aims to explore the various scenarios under which this global target could be met. We believe this revised focus more effectively aligns with the objectives and findings of our research, providing a clearer context for the policy challenges ahead. **(Please see lines 24-31)**

Comment 32. Notably, sales AMU data will NOT provide evidence of a causal relationship and advocating for the importance of such data (line 21) is missing the point. In line 21 (and elsewhere, e.g., lines 217-218), I would rather see that the authors advocate for accurate animal-level data on AMU in livestock (rather than AMU sales data) because, unlike AMU sales data, surveillance of animal-level data will allow causal inference between AMU and AMR.

- Reply: Thank you for this comment. We fully agree with your observation. We have adjusted the statement underscoring the importance of more precise livestock biomass denominators and accurate animal-level data on AMU to effectively inform and enhance stewardship strategies. **(Please see line 502)**

Comment 33. Figure 1, step 3: The world human population dataset was used as an exogenous variable to project the livestock population growth. How exactly was that done?

- Reply: Thank you for this comment. We have adjusted the methodology describing in more detail how population was incorporated in the specification of the livestock projection model. Below for easy reference a snap of the section

*If the series is stationary, we proceed with the specification of a VAR model as in (1), where y_t^{rl} is the vector of log transformed livestock numbers endogenous variables in subregion t and species l at time t Γ is a parameter matrix, C is the vector of coefficients for the constant/trend terms D_t^{rl} , β is the parameter matrix of the **human population** exogenous term, z_t^{rl} , and v_t^{rl} is a vector of stochastic error terms.*

$$y_t^{rl} = \Gamma y_{t-1}^{rl} + \dots + \Gamma_{p-1} y_{t-p+1}^{rl} + CD_t^{rl} + \beta z_t^{rl} + v_t^{rl} \quad (1)$$

Comment 34. Lines 66-68: Please list the species included in this section rather than in the Supplementary Material.

Reply: Thank you for this comment. We have included a more detailed description of the species consider in the study in methodology sections **(Please see line 73-74)**.

Comment 35. 13. Lines 87-89: Does this dataset include AMU both on- and extra-label?

Reply: Thank you for this comment. ANIMUSE data are mainly from sales and imports, so yes possible off-label use would also be covered by the data.

Comment 35. 14. Lines 100-101. The original PCU by the ESVAC of the EMA in the EU was developed to represent the animal's average weight at treatment in EU countries. It is unclear if, in this study, the authors used the original PCU methodology (developed and parametrized for the member countries of the EU) or if the authors have modified PCU to reflect animal weight at slaughter in different countries (see equations 7 and 8 in supplemental materials). The average weights calculated for this report are larger than the estimated weights at the time of treatment, resulting in a larger denominator and a decreased relative mg/kg estimate of antimicrobial agents intended for animal use. Consequently, the results in WOAHS analyses of antimicrobial quantities adjusted by animal biomass are not directly comparable to ESVAC or CIPARS estimates, which are based on treatment weights.

➤ Reply: Thank you for your comment. Yes, there are various PCU methodologies for determining average animal weights to calculate total biomass. The ESVAC uses estimated average weights at the time of treatment, CIPARS utilizes standard weights at the time of treatment. However, to the best of our knowledge, these conversion factors are primarily available for Europe, Japan, Canada, and a few other countries. Given the global scope of our study, estimating weights at the time of treatment is not feasible due to the lack of comprehensive data.

Kindly note that the livestock AMUQ projections in this study are based on the Livestock Biomass Conversion (LBC) method, rather than the PCU. The LBC is a new approach developed by FAO to calculate biomass, and it is presented for the first time in this publication. This methodological approach is one of the main contributions of our paper. We have provided a detailed description of both the LBC and PCU methodologies in the revised version of the methodology section for greater clarity and comparison.

Comments 36. The authors failed to explain why PCU consistently predicts less AMU than LBC. If PCU measures the weight at treatment while LBC reflects the weight at slaughter, then, in fact, PCU is a more correct reflection of the actual AMU use, albeit it requires knowledge of the weight at treatment for the livestock population in different (sub)regions. The differences in derivation and the estimated values for PCU and LBC must be explained. This is important because of considerable differences in estimates in some of the world's regions, and the reason for such differences needs to be explained.

- Reply Thank you for your question. We recognize the need to better explain the PCU and LBC methods in our methodology and to enhance the interpretation of our results. Please note that the livestock AMU projections in this study are based on the Livestock Biomass Conversion (LBC) method, not on the PCU method. The PCU tends to predict lower AMU because it assumes a higher biomass level by calculating it as the entire animal population per species times the average weight at slaughter age, effectively treating all animals as adults. In contrast, the LBC method, supported by FAO's internal GLEAM dataset, classifies animal species by commodity, production system, and cohort. This allows for a more accurate and consistently lower population weight, leading to more precise AMU projections. We hope this explanation clarifies our approach and improves the interpretation of our results. Please find below a detailed technical explanation of the two methods below.

PCU

The PCU is calculated in WOA reports, by multiplying the total number of animal species (i) in each country by its average live weight at the time of treatment, w_i , as (7). The expression in equation (1) was further standardized by Radke (2017) to account for the differences in animal weight and the number of production cycles as described in equation (2).

$$PCU_i = l_i \times w_i \quad (1)$$

$$PCU_{i,j} = l_{i,j}(1 + n_{i,j}) \times \left(\frac{Q_i}{R_k}\right) \quad (2)$$

where j denotes the production system of a country, $n_{i,j}$ is the number of production cycles for each species (i) in each production system (j). Q_i represents the total quantity of meat in each country for each animal species (i), and R_i denotes the carcass weight-to-live weight ratio for each animal species (i). As shown, the PCU method, as described in equation (2), relies on a single average carcass/slaughtered weight of animal species (i) in each production system (j). Therefore, this approach may result in skewed PCU measurements because it does not consider the weight of each animal at the specific cohort level. In other words, it assumes all the animals in the population to be adults.

LBC

To overcome this limitation, we introduced a new metric: the livestock biomass conversion (LBC) index. The LBC considers variations in livestock species, livestock commodities, production systems, cohorts, and live weight by cohort. This calculation was performed using the following equation:

$$LBC_{ic} = A_i \times n_i \times \left(\frac{L_{i,k}}{A_i}\right) \times \left(\frac{L_{i,p}}{L_{i,k}}\right) \times \left(\frac{L_{i,c}}{L_{i,p}}\right) \times W_{i,c} \quad (3)$$

where LBC_{ic} represents the livestock biomass indicator for animal species (i) in cohort (c); A_i is the total number of animal species (i); n_i denotes the number of production cycles used to account for lifespan differences among species (i), $L_{k,i}$ is the number of livestock species (i)

under commodity group (k); $L_{p,i}$ is the number of livestock species (i) under commodity group (k) and production system (p); $L_{c,i}$ is the number of livestock species (i) under commodity group (k), production system (p), and cohort (c); and $W_{c,i}$ is the cohort weight for animal species (i) under commodity group (k) and production system (p).

Finally, the LBC for animal species (i), accounting for all cohorts, is calculated by taking the linear summation of equation (4) as follows:

$$LBC_i = \sum_{c=1}^n LBC_{ic} = \sum_{c=1}^n \left(A_i \times n_i \times \left(\frac{L_{i,k}}{A_i} \right) \times \left(\frac{L_{i,p}}{L_{i,k}} \right) \times \left(\frac{L_{i,c}}{L_{i,p}} \right) \times W_{i,c} \right) \quad (4)$$

where LBC_i represents the livestock biomass indicator for animal species (i), and n denotes the cohort number.

Comment 37. 15. Lines 110-111. Did this sharp decline in AMU due to decreases in swine biomass from ASF have an appreciable impact on your projections for this region?

- Reply: Thank you for this comment. Evaluating whether ASF caused a structural change in the time series dynamics, and consequently in our projections, requires to be empirically tested. However, this requires more post-shock data observations. Nevertheless, it is noteworthy the capacity of the model in considering the speed of adjustment following the ASF exogenous shock and correcting for deviations from the long-run equilibrium. We have highlighted this point as a potential area for further research in the paper. Similar type of analysis has been previously conducted by Acosta et al. (2020) in their study on HPAI in Mexico (<https://onlinelibrary.wiley.com/doi/full/10.1111/1467-8489.12387>).

Comment 38. Lines 113-116. How is the 95% CI estimated? What does the 95% CI reflect? Is that the uncertainty in the projected livestock population growth or something else?

- Reply. Thank you for this comment. We have revised the paper to better explain the interpretation of the confidence intervals (CI). Yes, the 95% CI reflects the uncertainty in our projections, providing the range within which the actual values are expected to fall 95% of the time. These CIs are estimated by analyzing the residuals of the model and comparing the fitted values with the actual historical values. This analysis helps estimate the variance and covariances of the residuals, which in turn are used to calculate the forecast error variance. The CIs are then constructed using the point forecasts and the standard errors derived from the forecast error variances. We have also added this explanation in the methodology.

Comment 39. Lines 136-140. Please comment on whether this high proportion (73%) comes from the projected livestock growth (and/or projected human population growth), and/or projected increases in AMU. This concentration underscores the pivotal role these subregions play in the global AMU landscape within the livestock sector.” Related to the previous comment (Main Comment #1), it seems misleading to say that these subregions will play a pivotal role in AMU in the livestock

sector without acknowledging that the projected AMU growth reflects the projected animal biomass growth.

- Reply: Thank you for this comment. Please note that we have decided to prioritize the regional results in the paper to make room for incorporating the requested sensitivity analysis and robustness check. However, we have revised the statement to clarify that the projected AMU growth aligns with the projected animal biomass growth. Furthermore, we have highlighted that the regions experiencing the highest AMUQ growth are also those expected to significantly contribute to the global supply of animal-source foods, driven by increased food demand by 2040. **(Please see line 314)**

Comment 40. Lines 241-243. This is an overly simplistic view that ignores other factors that have nothing to do with AMU, and which can be expected to affect the trajectories, including human consumption preferences (different livestock species and production categories have different needs for AMU), livestock numbers, livestock health, and the availability of and access to methods to prevent infections in different regions.

- Reply: Thank you for this comment. We agree that the trajectories of AMU are influenced by a wider range of complex. We have expanded the discussion to better acknowledge these factors and their potential impacts on AMU projections. **(Please see line 468)**

Comment 41. Lines 244-253. The main unrecognized limitation is the quality (and potential bias in an unknown direction) of the used AMU sales data.

- Reply: Thank you for pointing out this critical issue. We have acknowledged and discuss extensively in the revised version that the use of AMU sales data as a proxy for AMU in livestock introduces several potential biases. Further as previously discussed to address the potential shortcomings of AMUQ sales data, we have conducted a robustness check using an alternative methodological approach for projecting AMUQ by 2040. **(Please see page 17 and line 478)**

Comment 42. Line 249-251. See above Main Comment #7 related to the inclusion of species and production settings for this aggregated dataset. Lines 648-649: “Our analysis encompasses primary livestock species, such as cattle, pigs, chicken, goats, and sheep.” Please specify exactly which species were considered in this analysis. Based on this and previous text, for example, it is unclear if AMU in horses is included in the analysis.

- Reply: Thank you for your comment. We appreciate the need for clarity regarding the species considered in our analysis. Specifically, our analysis includes AMU data for the following livestock species: cattle, pigs, chickens, goats, and sheep. Horses were not included in this analysis. We will revise the manuscript to explicitly list the species considered to ensure there is no ambiguity.

Comment 43. Lines 662-663. Does the “variability in livestock biomass” account for increases in mean bodyweight due to genetic selection for larger animals?

- Thank you for your comment. In our analysis, we did not specifically account for increases in mean bodyweight resulting from genetic selection. However, we do consider the transition from traditional to more intensive production systems, which leads to changes in livestock biomass.

Comment 44. Minor comment: mass and weight are used interchangeably, though mass would seem to be a more appropriate term.

- Reply: Thank you for pointing this out. We have harmonized the terminology throughout the manuscript, using 'biomass' and 'weight' only when the original method or researched cited employs these terms.

Reviewer 3

Comment 45. I co-reviewed this manuscript with one of the reviewers who provided the listed reports. This is part of the Nature Communications initiative to facilitate training in peer review and to provide appropriate recognition for Early Career Researchers who co-review manuscripts.

Thank you for co-reviewing the manuscript. We greatly appreciate your efforts to contribute to the peer-review of this paper.

Reviewer 4

Thank you for taking the time and effort to review our work. We have carefully addressed each of your comments.

Comment 45. The authors investigate an important area of antimicrobial resistance, the use of antibiotics in the livestock sector. The analysis is at a global scale and therefore has to make some compromises in terms of disaggregation of detail in terms of biology, geography and time. These limitations are not dealt with in the discussion or introduction, and I would have expected a need for:

- Reply: Thank you for this comment. We have added a dedicated Limitations section to the manuscript, where we discuss these shortcomings in detail. We believe this addition enhances the transparency of the study and addresses the concerns raised. **(Please see page 21)**

- antimicrobials used - the focus of the paper is antibiotics, not all antimicrobials, antibiotics are not homogenous have different importances in terms of their use in livestock and people, different half lives in the environment and their use generates different resistant profiles.

- Reply: Thank you for this comment. In response, we have replaced the term "antimicrobials" with "antibiotics" throughout the manuscript to reflect the correct focus of the study. **(Please see the new title of the paper)**

Additionally, we have added a statement in the Limitations section to address the point raised by the reviewer, highlighting that "among the quantities of antibiotics used for each drug class in this study, it is important to note that not all classes are considered equally important to human medicine, as indicated by the WHO (2024) classification of medically important antimicrobials"

While we acknowledge the differences in environmental half-lives between antibiotic classes and their resulting impact on resistance profiles, we believe a detailed discussion of these aspects falls outside the scope of this paper, which focuses on usage trends. This topic would be more appropriately addressed in studies specifically examining the environmental dimensions of AMR, particularly in aquatic ecosystems.

- There remains uncertainty of how important livestock derived AMR is to human health, there also remains uncertainty of the impacts on livestock production efficiency with reduction of antibiotic use in livestock, this has implications of efficiency of land and water use and hence possible impacts of the environment as well as impacts on consumer welfare.
- Reply: Thank you for this comment. We agree with the reviewer that the impact of reduced antibiotic use on livestock production efficiency, and the resulting implications for land, water use, and consumer welfare, are significant but uncertain. This uncertainty highlights the importance of conducting sensitivity analyses on productivity gains associated with antibiotic use. However, we would like to clarify that our study primarily focuses on projecting potential future trajectories for the quantities of antibiotics used in livestock. The broader environmental and welfare impacts of reducing antibiotic use, including effects on production efficiency and resource use, warrant a separate and more targeted analysis that goes beyond the scope of our current work.
- country level data is where we would begin to have some potential interventions questions that can be dealt with by jurisdiction, understandably the authors have not gone to country level, yet this is where power lies for change.
- Reply: Thank you for this comment. The referee is correct in noting that country-level data would allow for more targeted interventions, as it enables comparisons between national policies and antibiotic use trends. However, the majority of countries worldwide still do not report antibiotic usage at the national level, limiting the feasibility of such an analysis on a global scale. For this reason, we have relied on regionally aggregated data provided by WOAHP for our analysis. While country-level studies are valuable, the global nature of our study necessitated this broader approach. We acknowledge that as more countries improve their reporting, future studies will be able to delve deeper into national-level trends and policy impacts.

Comment 46. With regards the temporal issues the analysis presented is dependent on two parameters, the biomass of the animals and the level of antibiotic use per biomass unit. These parameters have a high degree of uncertainty:

- reliance on FAOSTAT data for populations and animal weights is fraught with difficulties when estimating biomass and the methods to estimate biomass are not decided upon antibiotic use per unit of biomass is not well collected and published even in high income countries
- Thank you for your comment. We fully recognize the challenges of relying solely on FAOSTAT data for populations and animal weights, which can indeed lead to difficulties in accurately estimating biomass. To address these limitations, we have taken significant steps in our analysis to improve the robustness of these estimates.

First, we employed FAO internal Global Livestock Environmental Assessment Model (GLEAM) dataset, which offers a more detailed and structured understanding of livestock populations. GLEAM captures species-specific information, production systems, and live weight dynamics, following the IPCC Tier 2 methodology. This framework provides a more refined characterization of livestock biomass compared to relying on FAOSTAT data alone, as it accounts for animal lifespans, production cycles, and cohort dynamics. Second, we developed the Livestock Biomass Conversion (LBC) method as a new alternative to traditional biomass estimation approaches. Unlike methods that use average slaughter weights, which can lead to over- or underestimation, the LBC method allows for a more granular assessment by factoring in cohort-specific live weights across different species and production systems. This method ensures a more accurate calculation of livestock biomass by considering variations in production cycles, species, and geographic regions. By using the LBC method, we aim to reduce the uncertainties associated with traditional biomass estimates and enhance the precision of our antimicrobial use projections.

- I would recommend that more time is spent on these elements in the discussion as the paper is a guide to what has happened and what might happen, whereas it probably is as useful in terms of setting out what data needs to be collected or what data collection needs to be improved to reduce the uncertainty of current estimates and to improve the predictions of the estimates generated.
- Thank you for your insightful comment. We have expanded the methodology section to discuss our approach in greater detail, particularly focusing on how we address the uncertainties surrounding livestock biomass and antimicrobial use estimates. Our revised paper highlights the integration of the GLEAM model and the development of the Livestock Biomass Conversion (LBC) method, both of which aim to reduce uncertainties by providing more granular and species-specific data. This allows for a more accurate projection of antimicrobial use across various scenarios.

Furthermore, we have extensively addressed these issues in the limitations section. We have outlined the key challenges related to data collection, such as reliance on aggregated data, gaps in global reporting, and the use of sales data as a proxy for actual antimicrobial use. The

limitations section also emphasizes the need for improved data collection systems, including species-specific reporting, real-time tracking of antimicrobial use, and better global coverage, to enhance the precision of current estimates and reduce the uncertainty in future projections. **(Please see line 502)**

Comment 47. I would also recommend the authors look at the following paper:

Schroback, P., Dennis, G., Li, Y., Mayberry, D., Shaw, A., Knight-Jones, T., Marsh, T.L., Pendell, D.L., Torgerson, P.R., Gilbert, W., Huntington, B., Raymond, K., Stacey, D.A., Bernardo, T., Bruce, M., McIntyre, K.M., Rushton, J., Herrero, M., 2023. Approximating the global economic (market) value of farmed animals. *Global Food Security* 39, 100722. <https://doi.org/10.1016/j.gfs.2023.100722>

- Reply: thank you for the reference. We have acknowledged the importance of accurately estimating livestock biomass, not only for refining AMUI metrics but also for assessing the economic value of farmed animals. **(Please see line 45)**

Comment 48: And if possible access the short studies that were carried out by OECD on antibiotic use in China and Brazil approximately 5 years ago. These had very large variations from the published data at the time.

- Reply: Thank you, the references have referenced it, in the discussion section, emphasizing the impact of stricter antibiotic regulations on AMUQ pathways in Brazil in the methods, and China's significant role in global AMU trends in the discussion. **(Please see line 248, and 438)**
 - OECD. (2019a). Antimicrobial use, resistance and economic benefits and costs to livestock producers in Brazil. OECD Publishing. <https://doi.org/10.1787/27137b1e-en>.
 - OECD. (2019b). Antibiotic use and antibiotic resistance in food-producing animals in China. OECD Publishing. <https://doi.org/10.1787/4adba8c1-en>.

Antimicrobial Use in Livestock: Future Global Perspectives

NCOMMS-24-36673

Response to the Second Round of Reviewers' Comments

December 2, 2024

1. Reviewer # 2.

We sincerely appreciate your positive feedback, recommendation for publication, and valuable insights that helped improve our manuscript.

- 1.1 Line 453-454: What is meant by (and what are the implications of) "coordinated effort that targets reductions in LBIO?" This might be misinterpreted as "the solution to rising livestock AMUQ is to cull the animals" to reduce the animal biomass. With growing human populations worldwide, the requirements for animal-derived proteins will likely increase. While studies that "optimize" the human diet in terms of global cropping capabilities (e.g. EAT-Lancet Report, 2019) suggest a decreased reliance on animal proteins, this is not as feasible in developing countries where LBIO is intimately tied to the well-being of people living below the poverty line. Is it appropriate or feasible to highlight this LBIO-reduction intervention, even if it is mathematically sound? To avoid misinterpretation, we recommend revising the statement or adding a disclaimer to clarify.

Reply: Reply: Thank you for raising this point. We have revised the text to clarify that optimizing LBIO focuses on improving livestock efficiency through better management and technological advancements, emphasizing productivity per animal rather than expanding herd sizes. The revision also highlights the importance of integrated strategies, especially in middle- and low-income countries where livestock supports livelihoods and food security. Hopefully, this ensures clarity and avoids misinterpretation. (Please see lines 453–460)

2. Reviewer #3 (Remarks to the Author):

Thank you for contributing to this review. We greatly value your time and effort in helping us improve the manuscript.

3. Reviewer #4 (Remarks to the Author):

We greatly appreciate your detailed and constructive feedback, which has significantly enhanced the clarity and robustness of our manuscript. We have addressed all your suggestions, including revising statements for accuracy, clarifying methodologies, and incorporating new references. Your comments have helped refine the discussion and highlight key areas for future research.

3.1 General comment, use of acronyms does not help the reader and some of the acronyms used are not common and, in some cases, not explained, for example LBIO, AMUI - do a search a replace and get rid of them.

- Reply: We appreciate your suggestion to replace acronyms with their full terms for improved clarity. We believe that removing acronyms such as AMUI, AMUQ, LBC, PCU, and LBIO would disrupt comprehension, as they are integral to the analysis. Kindly note that these acronyms are used throughout the equations, figures, and scenario definitions, and their removal could hinder the coherence of the paper. Nevertheless, to address your concern, we have ensured that these acronyms are clearly defined upon their first mention and consistently applied thereafter. We trust this approach provides an appropriate balance between clarity and readability.

3.2 Given you have decided to develop a different means of calculating biomass you should look at the GBADs technical guide (<https://animalhealthmetrics.org/gbads-technical-guide/>) which has a section on biomass estimations. And there should be a comparison between the biomass from the PCU and the LBC methods to show the differences.

- Reply: Thank you for your suggestion to review the GBADs technical guide and to compare methodologies. This has been a valuable opportunity for us to reflect further on the distinctions and complementarities between the approaches.

We have reviewed the GBADs technical guide on biomass estimations and the background paper by Li et al. (2024) and have cited this. Below, we present our observations:

- The Livestock Biomass Conversion (LBC) method introduced in this paper is a newly developed FAO approach for quantifying livestock biomass at the global level. While it builds on similar principles, it is distinct from the GBADs methodology.
- The GBADs methodology has proven highly effective for national and subnational analyses, as demonstrated in its detailed work in Ethiopia. However, its global estimates rely on a simpler approach, described as suitable for global-level applications (GBADs Technical Guide, p. 25).

- GBADs livestock biomass estimates, displayed in their dashboard, rely on FAO data and conversion factors (see GBADs dashboard metadata: <https://gbadske.org/dashboards/biomass/>). However, the LBC method leverages FAO’s internal GLEAM (Global Livestock Environmental Assessment Model) database, based on the TIR 2 approach, which allows for disaggregated estimations by species, cohort, production systems, and regions. This granularity provides a more nuanced and precise estimation of livestock biomass globally.
- While the GBADs approach excels at providing insights at a national and subnational level, the LBC method complements this by offering enhanced precision for global-level analyses. We have clarified this distinction in the text to highlight the complementary nature of both approaches. Thank you for encouraging this important discussion.

3.3 Figure 2 does not seem to make sense. The AMU per unit of biomass (which I think you refer to as AMUI) should be an input variable to the estimation of the quantity of AMU overall (which I think is referred to as AMUQ) not something that changes with the way biomass is calculated. Have I missed something in the calculations or is there something that needs more explanation?

- Reply: Thank you for raising this important question. We appreciate the opportunity to clarify the methodology. We have revised the methods and results sections to better explain how variations in livestock biomass (LBIO) estimation influence AMUI and emphasized the interconnection between these metrics. As
 - AMUI is calculated as the ratio of AMUQ (antimicrobial use quantity) to LBIO (livestock biomass). This is described on page 8; equation XIII as follows:

$$AMUI_r = \frac{AMUQ_{cr}}{LBIO_{cr}} \quad (xii)$$

- Here, AMUI is a derived measure that depends on the numerator (AMUQ) and the denominator (LBIO). The changes in AMUI observed in Figure 2 reflect differences in the calculation of LBIO using alternative methods. For example, the LBC method provides a more granular estimation of LBIO, accounting for species, production systems, and cohort-specific weights, which impacts the denominator and, consequently, the calculated AMUI.
- We have added further explanation in the text to address this and avoid potential confusion. Thank you for pointing this out.

3.4 There is a circularity of the estimation of the AMU per unit of biomass that does not feel very sound.

- **Reply:** Thank you for highlighting this concern. We understand how the relationship between AMUQ and AMUI might appear circular at first glance. However, as described above, AMUQ (numerator) is reported independently from biomass estimation (denominator). The LBIO is calculated separately using the LBC method and does not influence the AMUQ data. We have clarified this further in the text to ensure transparency.

3.5 Where is the use of reported AMU overall coming into the calculations?

- **Reply:** Thank you for raising this question. Reported AMU overall (AMUQ) serves as the numerator in determining antimicrobial use intensity (AMUI), as described in equation (xii). We have clarified this in the methods section to avoid ambiguity.

3.6 Has there been any attempt to relate the biomass in different production systems as well as different species. For example dairy versus beef in cattle or broiler versus layers in chickens. These then begin to unravel some of the critical questions on data

- **Reply:** Thank you for this suggestion. The LBC method accounts for variations in livestock biomass by species, commodity types, and production systems. However, the AMUQ data from the ANIMUSE dataset is aggregated across all livestock species and production systems, limiting our ability to disaggregate AMU by age, sex, or production system. We have emphasized this limitation in the discussion section as an area for improvement and a priority for future data collection efforts.

3.7 Line 525-26 - states that the biomass calculation used is more accurate than the PCU, what has been the gold standard to allow this statement to be made?

- **Reply:** Thank you for this observation. We recognize that there is no universally accepted “gold standard” for livestock biomass estimation. The LBC method provides greater granularity compared to the PCU approach, which uses average slaughter weights as a proxy for biomass. To avoid misinterpretation, we have revised the statement as follows:

“This study introduces the LBC method, providing a more granular assessment of livestock biomass at the global level. By capturing the complexity of livestock herd structures, the LBC method incorporates species-specific live weights, disaggregated by commodity groups, production systems, production cycles, and cohort-level details, offering a more detailed approach.”

3.8 Given that you have estimated biomass by species, production systems, age and sex it would make sense to present an estimate of the AMU broken down more carefully. If you cannot do this by age and sex then at least by production system. This is particularly important in the cattle and pigs which have the highest estimated use of antibiotics

Reply: We agree that this would provide valuable insights. However, as mentioned, the AMUQ data from the ANIMUSE dataset is aggregated across species and production systems, limiting our ability to provide such disaggregation. We have explicitly noted this limitation in the discussion and emphasized the need for more detailed data in future research.

3.9 No mention or discussion of the type of antibiotic and this is critical when thinking through the implications and the authors should be encouraged to look at the recent paper by Narmada et al (2024) <https://doi.org/10.1186/s12879-024-09847-3>

Reply: Thank you for highlighting this important point. In response, we have revised the manuscript to address the lack of antibiotic-class-specific data in the ANIMUSE dataset, noting its limitation in analyzing contributions of specific classes to AMUQ trends and resistance patterns.

3.10 line 36 "diving"?. Thank you. This should be "dividing" and has been corrected.

3.11 line 47 First spelt incorrectly. Thank you for pointing this out. We have reviewed the spelling throughout the paper.

3.12 line 244 - do you mean "developing countries"? should this not read low and middle income countries, or indeed should it not read high income countries. There is little evidence that livestock numbers are reducing in the LMICs, it is the reverse. The downward trend in high income countries have been a reduction in animal numbers but an increase in average animal size in the case of cattle and a faster turnover in the case of poultry and pigs

Reply: Thank you for your insightful comment. You are correct in noting that the trends in livestock numbers vary significantly between high-income and low- and middle-income countries. After careful consideration, we have decided to delete the statement regarding the reduction in animal numbers to avoid potential misinterpretation. This adjustment ensures that the discussion more accurately reflects the observed trends and nuances across different income groups and regions. We appreciate your attention to this detail.